# DO EGOCENTRIC VIDEO-LANGUAGE MODELS TRULY UNDERSTAND HAND-OBJECT INTERACTIONS?

**Boshen Xu**[1]   **Ziheng Wang**[1*]   **Yang Du**[1*]   **Zhinan Song**[1]
**Sipeng Zheng**[2]   **Qin Jin**[1†]
[1] Renmin University of China
[2] Beijing Academy of Artificial Intelligence

## ABSTRACT

Egocentric video-language pretraining is a crucial step in advancing the understanding of hand-object interactions in first-person scenarios. Despite successes on existing testbeds, we find that current EgoVLMs can be easily misled by simple modifications, such as changing the verbs or nouns in interaction descriptions, with models struggling to distinguish between these changes. This raises the question: "Do EgoVLMs truly understand hand-object interactions?" To address this question, we introduce a benchmark called **EgoHOIBench**, revealing the performance limitation of current egocentric models when confronted with such challenges. We attribute this performance gap to insufficient fine-grained supervision and the greater difficulty EgoVLMs experience in recognizing verbs compared to nouns. To tackle these issues, we propose a novel asymmetric contrastive objective named **EgoNCE++**. For the video-to-text objective, we enhance text supervision by generating negative captions using large language models or leveraging pretrained vocabulary for HOI-related word substitutions. For the text-to-video objective, we focus on preserving an object-centric feature space that clusters video representations based on shared nouns. Extensive experiments demonstrate that EgoNCE++ significantly enhances EgoHOI understanding, leading to improved performance across various EgoVLMs in tasks such as multi-instance retrieval, action recognition, and temporal understanding. Our code is available at https://github.com/xuboshen/EgoNCEpp.

## 1 INTRODUCTION

Humans have long envisioned embodied agents that can perform various societal roles. A promising approach to realizing this vision involves leveraging knowledge from egocentric demonstrations to train agents to imitate human actions during daily activities. Egocentric videos, captured from a first-person view using wearable devices, effectively showcase how individuals interact with nearby objects using their hands. This has sparked significant interest in understanding egocentric video, particularly hand-object interactions, due to its potential applications in VR/AR (Grauman et al., 2024a; Plizzari et al., 2024) and embodied agents (Zeng et al., 2023; Zheng et al., 2023).

Recent works (Lin et al., 2022) have utilized the large-scale dataset Ego4D (Grauman et al., 2022) to pretrain egocentric video-language models (EgoVLMs), enhancing performance in tasks, such as egocentric video-text retrieval (Lin et al., 2022; Sigurdsson et al., 2018b) and action recognition (Sigurdsson et al., 2018a). However, despite the impressive capabilities of these models and their benefit from large-scale pretraining on hand-object interaction (HOI) data, we have identified a critical issue in video-text matching: when tasked with selecting the correct sentence for a video from the sentences where the verb or noun varies significantly in meaning, EgoVLMs often fail to make accurate distinctions, as shown in Figure 1. This raises an important question: Do existing EgoVLMs truly understand egocentric hand-object interactions?

To delve deeper into this question, we introduce **EgoHOIBench**, a novel multi-choice testbed derived from Ego4D. This benchmark is specifically designed to assess the ability of EgoVLMs to comprehend

---

*Equal contribution.
†Corresponding author.

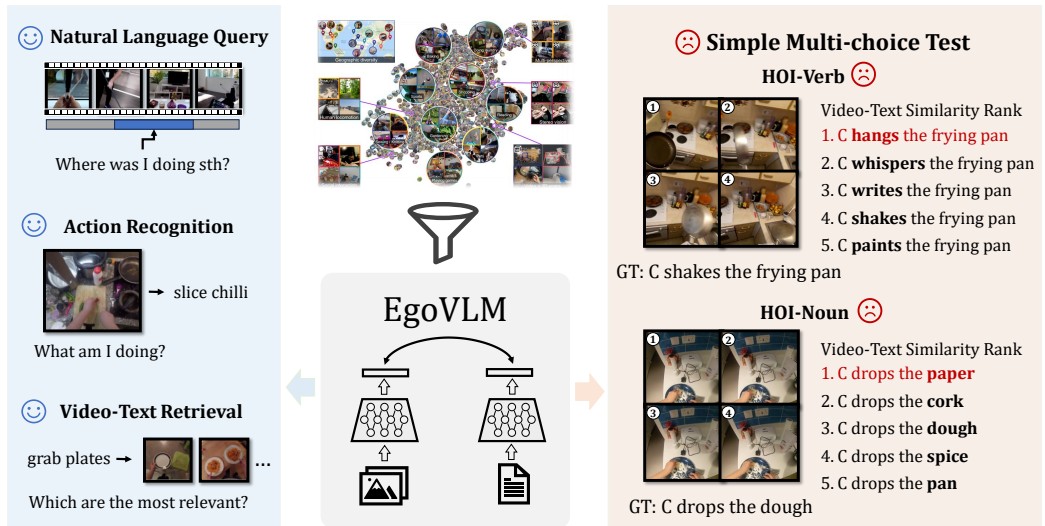

Figure 1: Although EgoVLMs have been pretrained on millions of worldwide egocentric videos and applied to challenging downstream tasks like video-text retrieval, we observe that they often fail to select the matched sentence from the simplest word-substituted candidates for videos.

HOI combinations with verbs or nouns varies through video-text matching. After evaluating state-of-the-art EgoVLMs on EgoHOIBench, we were surprised to observe a substantial decline in performance. Despite being trained on extensive EgoHOI data, these models still exhibit difficulty in accurately recognizing HOIs when confronted with even the most basic word substitutions.

We attribute this suboptimal performance of these models primarily to *a lack of fine-grained negative supervision* during the pretraining process. In egocentric video-language pretraining, which employs video-to-text and text-to-video contrastive losses (i.e., EgoNCE (Lin et al., 2022) and InfoNCE (Zhao et al., 2023)), training batches often contain many easy negative samples (e.g. "cut grass" vs. "pick apple"). While these negatives facilitate model generalization across HOI sentences with simultaneous verb-noun changes, they fail to provide effective supervision for understanding the nuances of HOI combinations, such as distinguishing between "shakes the frying pan" and "hangs the frying pan". Consequently, the models exhibit fragile robustness when evaluated on EgoHOIBench. One potential solution is to enhance fine-grained supervision through hard negative mining (Yuksekgonul et al., 2023). We therefore propose generating hard negatives that differ by a single word, HOI-related noun or verb. However, this approach carries risks: it may disrupt the understanding of other unchanged words, potentially reducing performance and limiting model generalization to other tasks (Momeni et al., 2023). Thus, a carefully designed training strategy is essential to address these challenges.

Furthermore, EgoVLMs demonstrate *stronger robustness towards recognizing nouns* through our analysis of EgoHOIBench performance on HOI-verbs and HOI-nouns. By visualizing video representations in a low-dimensional space, we reveal that these EgoVLMs develop object-centric feature spaces, where representations with the same nouns are more robustly encoded and clustered than those with the same verbs. This phenomenon, leading to improved performance on HOI-nouns, can be viewed as advantageous, as previous studies have shown the effectiveness of establishing object-centric features through additional structures trained on object images (Escorcia et al., 2022) or supervision from HOI detection tasks (Zhang et al., 2023; Li et al., 2021b).

In this work, we aim to preserve the object-centric nature of the feature space, without requiring additional visual data or architectural changes, while simultaneously enhancing HOIw comprehension, from a contrastive learning perspective. To this end, we introduce **EgoNCE++**, a novel contrastive learning objective that incorporates asymmetric video-to-text and text-to-video losses. Specifically, the video-to-text loss enables the model to capture both word- and sentence-level semantics for each video through hard negative supervision, enhanced by generating HOI-related negative captions using large language models (LLMs) or leveraging vocabulary prior knowledge from the pretraining dataset. Conversely, the text-to-video loss preserves the established object-centric feature space by clustering

video representations with similar nouns in their captions. We conduct extensive experiments across various EgoHOI downstream benchmarks, demonstrating that EgoNCE++ significantly improves the generalization of EgoVLMs to other tasks in a zero-shot manner.

Our contributions in this work are threefold: (1) We develop **EgoHOIBench**, a novel benchmark specifically designed to evaluate EgoVLMs' capabilities in understanding variations of HOI combination. (2) We propose **EgoNCE++**, an innovative HOI-aware asymmetric contrastive learning objective for egocentric video-language pretraining. (3) Our experimental results demonstrate the versatility and efficacy of EgoNCE++, notably enhancing performance across three EgoVLMs and improving generalization on seven downstream EgoHOI tasks.

## 2 EGOHOIBENCH: DO EGOVLMS TRULY UNDERSTAND HOIS?

Existing benchmarks in egocentric vision have primarily focused on EgoHOI. For instance, Damen et al. (2021) emphasize action recognition in kitchen scenarios, which limits its ability to evaluate the broader knowledge embedded in VLMs. Wang et al. (2023b) propose assessing a model's temporal understanding by requiring it to distinguish between actions with similar semantics. Lin et al. (2022) suggest querying the correct video from multiple options across various scenarios based on provided text descriptions. In contrast to these testbeds, EgoHOIBench introduces a straightforward multi-choice test for video-to-text matching, featuring comprehensive real-world scenarios and a rich, diverse vocabulary centered on hand-object interactions (HOIs). This benchmark is designed to evaluate the ability of EgoVLMs to select the correct sentence from multiple HOI-related options using video-text matching.

### 2.1 NEW BENCHMARK FOR NUANCED EGOHOI DISTINCTION

To clarify the task definition, we design each EgoHOI multi-choice trial as follows: given a video segment $x$, the model is required to distinguish the correct caption $S^*$ from $N$ verb-focused hard negative captions $\{S_i\}_{i=1}^N$, where $S_i$ and $S^*$ differ only in the verb (HOI-verb task). Similarly, the model must identify $S^*$ from $N$ noun-focused hard negative captions $\{S_j\}_{j=1}^N$ where $S_j$ is generated by replacing the noun in $S^*$ with alternative nouns (HOI-noun task). A trial is considered successful only when the model accurately identifies the correct caption $S^*$ for both the HOI-verb and HOI-noun tasks.

We contend that an ideal EgoVLM should excel at solving these relatively straightforward tasks, provided that the choice options are not deliberately made excessively difficult. Therefore, leveraging LLMs to generate such less challenging options is a practical approach, given their robust world knowl-

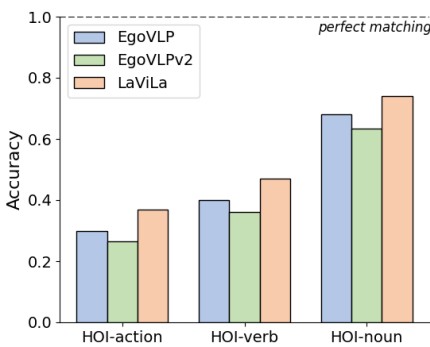

Figure 2: EgoHOI performance of EgoVLMs on EgoHOIBench.

edge and instruction-following capabilities. This method provides a more efficient and scalable alternative to relying on human labor. The LLM is prompted to perform word substitutions, randomly replacing HOI-related verbs or nouns with alternatives of different meanings. This process transforms the multi-choice candidates into alternative sentences, significantly altering their semantics. Ultimately, EgoHOIBench provides a comprehensive evaluation of models' understanding of EgoHOIs across 29K test trials. More details on the data construction process are available in Appendix B.

We evaluate three state-of-the-art EgoVLMs: EgoVLP (Lin et al., 2022), EgoVLPv2 (Pramanick et al., 2023), and LaViLa (Zhao et al., 2023). Surprisingly, all models perform poorly on EgoHOIBench, as illustrated in Fig. 2. To better understand the underlying reasons for this suboptimal performance, we focus on addressing two key questions:

- Why EgoVLMs struggle with the seemingly simple multi-choice test? (Section 2.2)
- Why performance on HOI-noun is better than that on HOI-verb? (Section 2.3)

## 2.2 Limitations of Existing EgoVLP Objective

Egocentric video-language pretraining (VLP) follows the standard VLP paradigm, which utilizes a dual-encoder architecture to perform contrastive learning between video and text modalities. Current EgoVLPs consider two symmetric contrastive learning objectives: InfoNCE (Zhao et al., 2023) and EgoNCE (Lin et al., 2022; Pramanick et al., 2023; Zhang et al., 2023; Phan et al., 2024).

**InfoNCE** (Radford et al., 2021). InfoNCE is a widely used contrastive learning objective that encourages positive video-text pairs closer while pushing negative pairs further apart through an online cross-entropy loss. The symmetric InfoNCE loss, applied to a batch of (video, caption) samples, can be formulated as:

$$\mathcal{L}^{\text{info}} = -\frac{1}{B}\Big(\sum_{v_i \in \mathcal{B}(v)} \log \frac{\exp(v_i \cdot t_i/\tau)}{\Sigma_{t_j \in \mathcal{B}(t)} \exp(v_i \cdot t_j/\tau)} + \sum_{t_i \in \mathcal{B}(t)} \log \frac{\exp(t_i \cdot v_i/\tau)}{\Sigma_{v_j \in \mathcal{B}(v)} \exp(t_i \cdot v_j/\tau)}\Big) \quad (1)$$

where $(v_i, t_i)$ denotes the $L_2$ normalized feature vectors of the $i$-th (video, caption) sample within a batch. $\mathcal{B}(v) = \{v_i\}_{i=1}^{B}$ and $\mathcal{B}(t) = \{t_i\}_{i=1}^{B}$ refer to the videos and captions of the batch $\mathcal{B} = \{(v_i, t_i)\}_{i=1}^{B}$, respectively.

**EgoNCE** (Lin et al., 2022). It is specifically tailored for egocentric scenarios. As shown in the following video-to-text loss, EgoNCE enhances the learning of subtle differences in scenes by enlarging the batch to include additional video clips with visually similar backgrounds. It also expands positive video-text pairs by including texts that depict similar HOIs occurring in different contexts. The video-to-text loss is defined as:

$$\mathcal{L}_{v2t}^{\text{ego}} = -\frac{1}{2B} \sum_{v_i \in \widetilde{\mathcal{B}}(v) \cup \mathcal{B}(v)} \log \frac{\sum_{t_k \in \mathcal{P}(t_i)} \exp(v_i \cdot t_k)}{\sum_{t_j \in \mathcal{B}(t)} \exp(v_i \cdot t_j) + \sum_{t_{j'} \in \widetilde{\mathcal{B}}(t)} \exp(v_i \cdot t_{j'})} \quad (2)$$

where $\widetilde{\mathcal{B}} = \{(v_{i'}, t_{i'})\}_{i'=1}^{B}$, $\widetilde{\mathcal{B}}(v) = \{v_{i'}\}_{i'=1}^{B}$ and $\widetilde{\mathcal{B}}(t) = \{t_{i'}\}_{i'=1}^{B}$ represent the enlarged batch samples. Each $(v_{i'}, t_{i'})$ corresponds to the (video, caption) pair sourced from the same recording environments as the $i$-th video clip in the original batch. Furthermore, $\mathcal{P}(t_i) \subseteq \widetilde{\mathcal{B}}(t) \cup \mathcal{B}(t)$ defines the set of captions, each containing at least one verb or noun that matches those in $t_i$. To save space, we omit displaying the text-to-video loss as it is symmetrically formulated.

**Lack of Fine-Grained Text Supervision for HOI-action.** While EgoVLMs pretrained with InfoNCE and EgoNCE have gained substantial knowledge about EgoHOI, they lack *fine-grained text supervision*. Specifically, InfoNCE often samples easy negative pairs (e.g., 'opens a drawer" vs. "picks an egg") without employing effective hard negative mining for text. Additionally, EgoNCE's positive sampling expansion strategy treats pairs like "opens a drawer" and "closes a drawer", or "opens a drawer" and "opens a bottle", as positive pairs, which weakens the model's understanding of fine-grained HOIs. As a result, these objectives often distinguish EgoHOIs based on simultaneous verb-noun variation, ignoring the need to learn the true semantics of HOI combinations. While these pretraining objectives are proven to be effective, we believe that a more generalizable EgoVLM should be capable of recognizing word-level variations in sentences.

## 2.3 Observation of Object-Centric Feature Space in EgoVLMs

**EgoVLMs Establish an Object-Centric Feature Space for HOI-noun.** While performance on the HOI-noun task shows room for improvement, results are even lower on the HOI-verb task. Recognizing complex actions and temporal dynamics is generally more challenging than identifying static objects (Damen et al., 2021). We hypothesize that this pattern extends to video-text matching, where matching a video to the correct verb is more difficult than matching it to the appropriate noun.

To test this hypothesis, we visualize egocentric video and text embeddings in a low-dimensional space to examine their distribution. Specifically, for visualizing verb-anchored text and video embeddings, we select several HOI-related verbs as anchors (e.g., "pick"). For each anchor verb, we gather 150 text captions that share the same verb but feature different nouns (e.g., "... pick apples ..." "... pick clothes ..." "... pick books ..."). We then visualize the embeddings of these verb-anchored texts and the embeddings of their paired videos. Similarly, we create visualizations for noun-anchored text and video embeddings. Using t-SNE (van der Maaten & Hinton, 2008) for dimensionality reduction, we generate visualizations, as shown in Figure 3, focusing on LaViLa's feature space. After egocentric

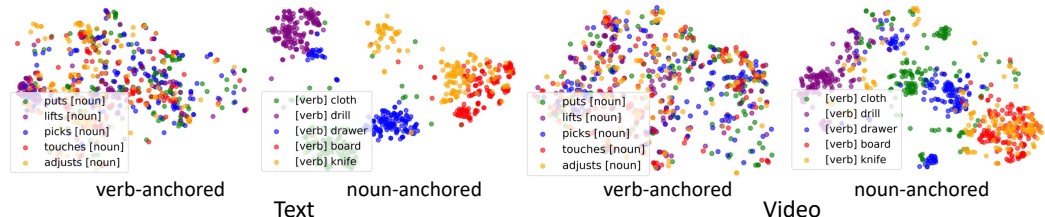

Figure 3: Visualization of LaViLa's feature space. Both video and text feature space exhibits the object-centric property. Apparently, the videos/texts are more separable by nouns, indicating a video is more easily matched with the correct noun on HOI-noun tests rather than verbs.

pretraining, the feature space reveals that noun-anchored embeddings form tighter clusters, while verb-anchored embeddings are more dispersed. This suggests that video-noun matching is easier than video-verb matching, which explains the poorer performance on HOI-verb tasks compared to HOI-noun tasks. A similar pattern is observed in EgoVLP's feature space, as detailed in Figure 12.

**Current Objectives Are Not Tailored for Learning Object-Centric Features.** Upon revisiting the InfoNCE and EgoNCE, we observe that the video and text embeddings generated during training tend to cluster around nouns. However, these objectives are not explicitly designed to learn an object-centric feature space. On the other hand, prior research (Zhang et al., 2023; Escorcia et al., 2022; Zhou et al., 2023) has demonstrated the benefits of enhancing object-centric features. Building on this insight, we aim to further strengthen these features through a contrastive learning perspective on the text-to-video side, thereby retaining the advantage for the HOI-noun task.

## 3 EGONCE++: HOI-AWARE ASYMMETRIC PRETRAINING OBJECTIVE

Building upon the analyses above, our primary goal is to enhance the model's sensitivity to word variations that benefits HOI-action recognition, while also reinforcing the object-centric feature space in EgoVLMs to maintain their advantage in HOI-noun tasks. To achieve this, we introduce a new contrastive learning objective called EgoNCE++, which incorporates asymmetric video-to-text (Section 3.1) and text-to-video losses (Section 3.2). The video-to-text loss enables the model to better understand HOI combinations by generating negatives through HOI-related word changes, while the text-to-video loss preserves object-centric feature properties by clustering video representations based on similar nouns. Figure 4 illustrates an overview of our method.

### 3.1 V2T: HOI-AWARE NEGATIVE GENERATION BY LLM OR VOCABULARY

To build a more robust EgoVLM that is sensitive to variations in HOI combinations, we focus on enhancing the negative text supervision by generating fine-grained hard negatives $\mathcal{N}(t)$ through targeted word changes to specific verbs or nouns. This approach ensures that each video $v$ is paired with a fixed set of nuanced hard negatives, while retaining easier negatives in $\mathcal{B}(t)$. The generated false HOI combinations $\mathcal{N}(t)$ provide more stable and high-quality supervision for understanding true HOI combinations compared to easy negatives in $\mathcal{B}(t)$.

We propose two methods for generating these hard negatives: (1) utilizing the vocabulary from the pretraining dataset, or (2) leveraging an LLM when the vocabulary is unavailable. The simplest approach involves substituting HOI-related words from the pretraining vocabulary, encouraging EgoVLM to better capture HOIs within the pretraining data. Specifically, we use spaCy (Honnibal et al., 2020) to extract all verbs and nouns in the pretraining dataset. Then, we replace HOI-nouns or HOI-verbs by extracted words randomly. When the pretraining vocabulary is insufficient or unavailable, an LLM can be employed to generate negatives. We prompt an LLM to use the JSON format for response and an in-context learning example to generate sentences that differ semantically from the original text. LLM-generated negatives are not only more fluent and diverse but also more aligned with real-world contexts, owing to the LLM's extensive world knowledge. Comparing these two strategies, negatives generated by the vocabulary help develop a better understanding within the pretraining dataset, while the LLM-generated negatives may generalize to unseen HOI combinations.

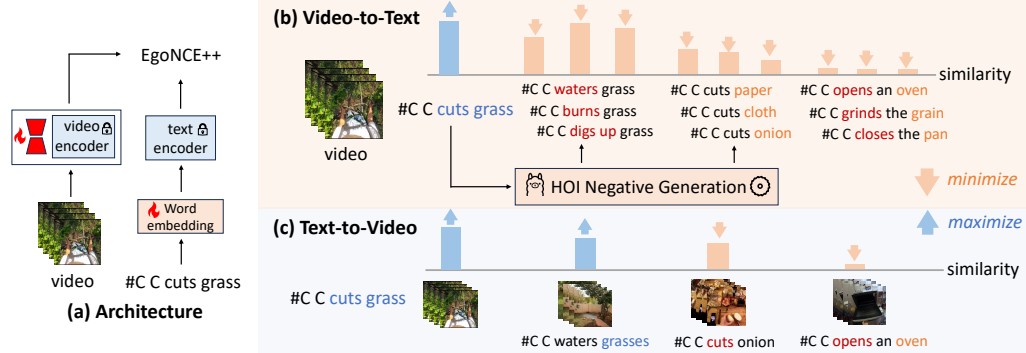

Figure 4: Illustration of our pretraining framework. (a) EgoVLMs are trained with EgoNCE++, where the visual encoder is trained using LoRA (Hu et al., 2022) to enhance video representation, while the text encoder remains frozen. Specifically, EgoNCE++ consists of (b) V2T: generating HOI-related negative captions for fine-grained supervision, and (c) T2V: strengthening the strong ability of EgoVLMs to recognize nouns by aggregating video features associated with similar nouns.

After generating negative captions with plausible semantics for videos, we apply the following supervision loss to improve HOI understanding from the video-to-text perspective:

$$\mathcal{L}_{v2t} = \frac{1}{B} \sum_{v_i \in \mathcal{B}(v)} \log \frac{\exp(v_i \cdot t_i)}{\Sigma_{t_j \in \mathcal{B}(t)} \exp(v_i \cdot t_j) + \Sigma_{t_k \in \mathcal{N}_{\text{noun}}(t_i) \cup \mathcal{N}_{\text{verb}}(t_i)} \exp(v_i \cdot t_k)} \tag{3}$$

where $\mathcal{N}_{\text{verb}}(t_i)$ and $\mathcal{N}_{\text{noun}}(t_i)$ denote the verb negatives and noun negatives, respectively. Our video-to-text loss guides videos to correct HOI meanings with supervision from both the coarse-grained easy negatives and fine-grained word substituted negatives.

### 3.2 T2V: OBJECT-CENTRIC POSITIVE VIDEO SAMPLING

Since our V2T negative mining on HOI-verbs might damage the strong recognition of nouns, we aim to maintain the noun clustering nature by T2V positive sampling on nouns. The text-to-video loss is designed to preserve the object-centric video features that enhance video-text matching. As discussed in Section 2.3, it is natural to reach the solution that we can continue to group video representations with the same nouns for given narrations.

To this end, we devise an object-centric text-to-video loss, where $\mathcal{P}_{\text{noun}}(v_i)$ denotes the videos that feature similar nouns in their captions:

$$\mathcal{L}_{t2v} = \frac{1}{B} \sum_{t_i \in \mathcal{B}(t)} \log \frac{\Sigma_{k \in \mathcal{P}_{\text{noun}}(v_i)} \exp(t_i \cdot v_k)}{\Sigma_{v_j \in \mathcal{B}(v)} \exp(t_i \cdot v_j)} \tag{4}$$

Different from EgoNCE which considers videos with either similar verbs or nouns as positives, we argue that only grouping videos with the same nouns is more suitable for learning the object-centric nature of EgoVLMs' feature spaces.

### 3.3 TRAINING STRATEGY

To refine the video representation of EgoVLMs for better generalization, we freeze the text encoder, except for the word embedding to adapt for novel negative sentence distribution, while fine-tuning the visual encoder using LoRA (Hu et al., 2022). The dual encoders are trained with both video-to-text loss by fine-grained negative text supervision and text-to-video loss with object-centric positive video sampling. Our final objective comprises the sum of text-to-video and video-to-text losses:

$$\mathcal{L}_{\text{EgoNCE++}} = \mathcal{L}_{t2v} + \mathcal{L}_{v2t} \tag{5}$$

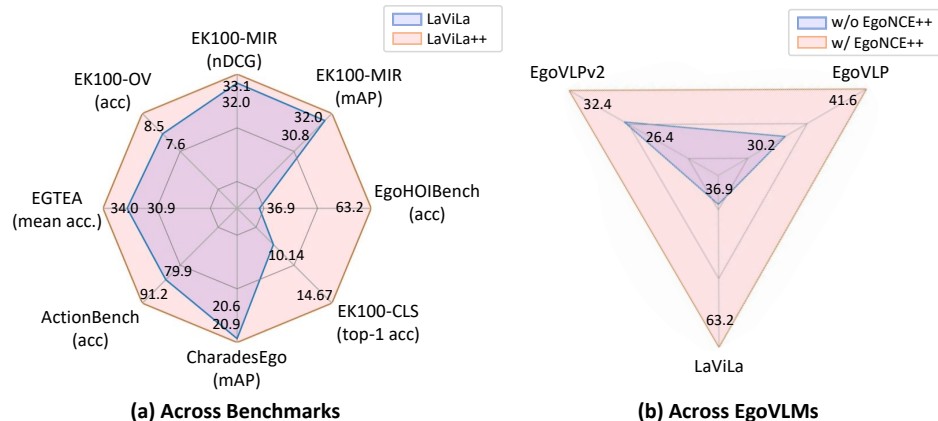

**(a) Across Benchmarks**     **(b) Across EgoVLMs**

Figure 5: Overview of experimental results. (a) LaViLa++ that is pretrained on LaViLa using EgoNCE++ achieves remarkable improvements across benchmarks under zero-shot settings, meanwhile (b) EgoNCE++ universally enhances HOI comprehension on EgoHOIBench across EgoVLMs.

# 4 EXPERIMENTS

## 4.1 EXPERIMENTAL SETTINGS

To ensure the robustness of our approach, we evaluate a range of well-known EgoVLMs including EgoVLP (Lin et al., 2022), EgoVLPv2 (Pramanick et al., 2023) and LaViLa (Zhao et al., 2023). Details of these models can be found in our Appendix C.1. In this paper, we continue to pretrain these models instead of training them from scratch due to computational resource constraints.

**Pretraining Dataset and Details.** Our pretraining video clips are sourced from EgoClip-3.8M (Lin et al., 2022), ensuring no overlap with the clips used in EgoHOIBench. The dataset focuses on EgoHOIs, excluding videos that primarily capture the activities of other persons, resulting in a dataset of 2.5 million entries. The videos are typically about 1 second long, accompanied by captions describing verbs and nouns relevant to hand-object interactions. During pretraining, we sample 4 frames from each video. We employ LoRA tuning with both rank and alpha set to 16. The models are continually pretrained for 10 epochs over a period of 12 hours using $8\times$ A800 GPUs, with a total batch size of 576. We utilize LLaMA3-8B (AI, 2024) to generate negative captions for the videos.

**Downstream Benchmark and Evaluation Setups.** We evaluate our model on three types of tasks across seven benchmarks in a zero-shot setting: (1) Open-vocabulary recognition: tasks that test video-text matching for video-and-language models. We evaluate on EgoHOIBench, EK-100-OV (Chatterjee et al., 2024), and ActionBench (Wang et al., 2023b). EgoHOIBench assesses models' sensitivity to HOI word changes, EK-100-OV evaluates the recognition of unseen object categories in kitchen scenarios, and ActionBench focuses on temporal understanding in open-world scenarios. (2) Multi-instance retrieval: conducted on Epic-Kitchens-100 (Damen et al., 2021), a kitchen-oriented retrieval benchmark where multiple video clips can correspond to the same narration. (3) Action recognition: tested on CharadesEgo (Sigurdsson et al., 2018a), EK-100-CLS (Damen et al., 2021), and EGTEA (Li et al., 2018). CharadesEgo presents an out-of-domain challenge (Lin et al., 2022; Zhao et al., 2023) for models trained on Ego4D with 157 indoor activity classes. EGTEA requires classifying 106 cooking activities, while EK-100-CLS evaluates 97 verbs and 300 nouns in kitchens.

## 4.2 MAIN RESULTS

**EgoNCE++ Enhances EgoVLMs' Sensitivity on HOI-related Word Changes.** On EgoHOIBench, models are required to comprehend HOI combinations and distinguish specific word changes in sentence candidates. As shown in Figure 5 (b), all EgoVLMs pretrained with EgoNCE++ exhibit significant improvements, showcasing the versatility of our method across various architectures, training strategies, and loss functions. The notable enhancements primarily arise from improved verb

Table 1: Comparison with state-of-the-art methods HelpingHands (Zhang et al., 2023) and HENASY (Phan et al., 2024) on zero-shot EK100-MIR and EGTEA. All these models are built upon LaViLa-Base model. The numbers of the method with * are sourced from Phan et al. (2024).

| METHOD | Extra Param | Epic-Kitchens-100-MIR | | | | | | EGTEA | |
| | | mAP (%) | | | nDCG (%) | | | | |
| | | V→T | T→V | Avg. | V→T | T→V | Avg. | mean-acc | top1-acc |
|---|---|---|---|---|---|---|---|---|---|
| LaViLa | - | 35.1 | 26.6 | 30.8 | 33.7 | 30.4 | 32.0 | 30.9 | 35.1 |
| HelpingHands* | 37M | 35.6 | 26.8 | 31.2 | 34.7 | **31.7** | **33.2** | 29.4 | 35.3 |
| HENASY* | 112M | 35.5 | 27.1 | 31.3 | 34.6 | **31.7** | 33.2 | 29.6 | **35.9** |
| LaViLa++ | 27M | **35.8** | **27.9** | **32.0** | **34.8** | 31.4 | 33.1 | **34.0** | 35.4 |

Table 2: Comparison of temporal understanding action recognition on ActionBench, where * denotes the fine-tuned model by Wang et al. (2023b).

| MODEL | InternVideo* | Clip-Vip* | Singularity* | Human | LaViLa | LaViLa++ |
|---|---|---|---|---|---|---|
| ACTION ACCURACY | 90.1 | 89.3 | 83.8 | 92.0 | 79.89 | 91.18 |

understanding, e.g. a **+34.02%** increase in verb accuracy leading to a **+26.32%** improvement in action accuracy for LaViLa++. Detailed numbers are provided in the Appendix C.3.

**EgoNCE++ Consistently Benefits EgoVLMs Across Multiple Benchmarks.** Taking a state-of-the-art EgoVLM LaViLa as an example as shown in Figure 5 (a), EgoNCE++ demonstrates consistent improvements across all benchmarks. From the perspective of video-text alignment during pretraining, EgoVLMs clearly benefit from EgoNCE++, leading to significant gains in HOI comprehension, especially reflected on Ego4D benchmarks including EgoHOIBench (**+26.32%**), ActionBench (**+11.3%**). Moreover, EgoNCE++ exhibits strong generalizations across other datasets, showing improvements on EK100-CLS (**+4.53%**), EK100-OV (**+0.9%**), and EGTEA (**+3.1%**)

**LaViLa++ Competes with SoTA Models in Zero-Shot Multi-Instance Retrieval and Action Recognition.** As shown in Table 1, LaViLa++ remains competitive with state-of-the-art models built upon LaViLa across all metrics for retrieval and action recognition tasks. Specifically, it achieves a notable **+1.2%** increase in average mAP over LaViLa, surpassing models that incorporate additional HOI detection supervision (Zhang et al., 2023) or hierarchical architecture (Phan et al., 2024). It also improves nDCG by **+1.1%** over LaViLa, achieving competitive results compared to other models. Furthermore, LaViLa++ demonstrates a significant **+4.4%** boost than other methods in mean accuracy on EGTEA, indicating that EgoNCE++ serves as a promising pretraining objective for EgoVLMs.

**EgoNCE++ Boosts Model Temporal Understanding Capability.** ActionBench (Chatterjee et al., 2024) focuses on temporal understanding tasks, such as distinguishing between "pick up" and "put down". In Table 2, we evaluate both LaViLa and LaViLa++ in a zero-shot setting. Although we do not specifically create negatives for temporal understanding, the results indicate that LaViLa++ can accurately classify verbs by distinguishing them from their antonyms. Our model surpasses the previous best models reported in Wang et al. (2023b) and even approaches human-level performance.

## 4.3 ABLATION STUDY

All ablation studies are conducted by pretraining the EgoVLP model (Lin et al., 2022). More detailed ablation studies can be found in Appendix C.4.

**Our Asymmetric V2T and T2V Losses Bring Collaborative Enhancement in Performance.** As shown in Table 3, our video-to-text supervision ("ours") significantly enhances the EgoVLM's ability to capture fine-grained details, achieving a +9.8% improvement in HOI-action and a +0.3% increase in mAP for generalization to EK-100-MIR, outperforming both InfoNCE and EgoNCE. Comparing the 3rd and 4th rows, we observe that combining our asymmetric video-to-text loss with text-to-video loss further strengthens the model's HOI comprehension, resulting in an additional +1.32% improvement in HOI-action on EgoHOIBench and enhanced generalization on EK100-MIR, with an additional +0.3% increase in mAP.

Table 3: Ablation of V2T and T2V losses.

| V2T | T2V | EgoHOIBench | | | EK-100-MIR | |
|---|---|---|---|---|---|---|
| | | verb | noun | action | avg.mAP | avg.nDCG |
| EgoNCE | EgoNCE | 40.27 | 68.60 | 30.16 | 22.2 | 26.7 |
| InfoNCE | InfoNCE | 40.70 | 68.86 | 30.51 | 22.1 | 26.5 |
| InfoNCE | EgoNCE++ | 40.60 | 69.15 | 30.62 | 22.3 | 26.7 |
| EgoNCE++ | InfoNCE | 54.56 | 68.96 | 40.31 | 22.4 | 26.9 |
| EgoNCE++ | EgoNCE++ | 56.11 | 69.05 | 41.63 | 22.7 | 27.1 |

Table 4: Ablation of the negative generator.

| GENERATOR | EgoHOIBench | | | EK-100-MIR | |
|---|---|---|---|---|---|
| | verb | noun | action | avg.mAP | avg.nDCG |
| none | 40.27 | 68.60 | 30.16 | 22.2 | 26.7 |
| rule-based | 43.52 | 68.94 | 32.63 | 22.1 | 26.7 |
| vocab-based | 54.46 | 68.56 | 40.07 | 22.5 | 27.1 |
| LLM-based | 56.11 | 69.05 | 41.63 | 22.7 | 27.1 |

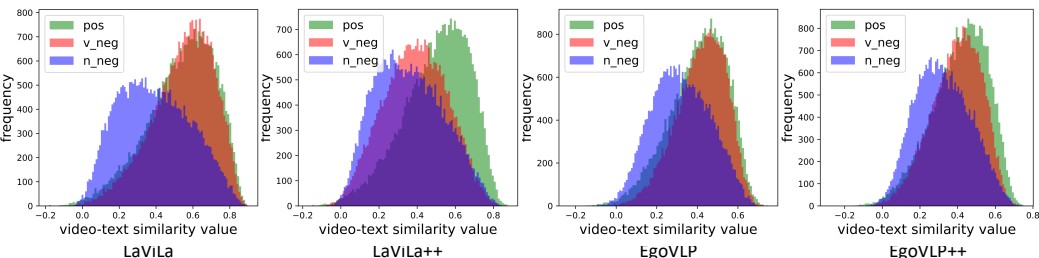

Figure 6: Histogram of video-text similarities for EgoVLP and LaViLa on EgoHOIBench. After applying EgoNCE++, the video-verb negatives are especially suppressed and thus the video-positives are more distinguished. LaViLa that is pretrained with InfoNCE benefits more from EgoNCE++ than EgoVLP that is pretrained with EgoNCE.

**Our Hard Negative Generation Performs Better than Rule-Based Generation.** We compare our LLM-based and vocab-based hard negative generation methods with the rule-based method: (1) LLM-based, where an LLM performs word substitutions through in-context learning; (2) vocab-based, where HOI verbs are replaced with arbitrary verbs from the predefined Ego4D vocabulary containing thousands of words. (3) rule-based, where hard negatives are selected by choosing captions with the highest BLEU (Papineni et al., 2002) scores from the sentences; As shown in Table 4, both the LLM-based and vocab-based methods result in significant improvements, surpassing the rule-based method by at least +7.44% on HOI-action and +0.4% on EK100-MIR in mAP. The rule-based method provides negatives that have already been encountered during pretraining, making it less effective. While the vocab-based method occasionally generates meaningless

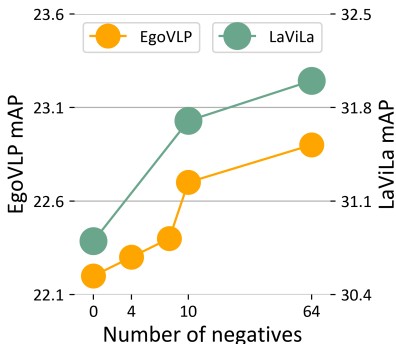

Figure 7: Scaling effect of negative number on EK-100-MIR (mAP).

HOI combinations, the LLM-based method produces more effective hard negatives, resulting in slightly better performance. The trade-off between the efficiency and efficacy of the vocab-based and LLM-based methods is demonstrated in Appendix C.5.

**Performance Improves as the Negative Number Increases.** Figure 7 illustrates the trend in mAP for EK-100-MIR as the number of negative samples increases. A clear correlation is observed: with more negatives leading to improved performance across various EgoVLM, such as EgoVLP and LaViLa. More negatives during pretraining significantly enhance the distinguishability of true video-HOI matches from other false HOIs, contributing to better performance.

### 4.4 FURTHER ANALYSIS

**Histogram of Video-Text Similarity on EgoHOIBench.** To examine how video-text similarities are changed by EgoNCE++, we visualize histograms of video-text similarities on EgoVLP and LaViLa in Figure 6. Video-positives roughly remain a high range of similarity, while the video-negatives are suppressed lower after applying EgoNCE++. We note that the LaViLa pretrained on InfoNCE benefits more from EgoNCE++ than EgoVLP pretrained on EgoNCE.

**Qualitative results.** For detailed qualitative results, please refer to Appendix D.

## 5 RELATED WORK

**Egocentric Hand-Object Interaction.** Captured by head-mounted cameras, egocentric hand-object interaction (EgoHOI) (Grauman et al., 2022; Chatterjee et al., 2024; Xue & Grauman, 2024; Plizzari et al., 2023; Zhang et al., 2022; Mangalam et al., 2024) provides insight into how humans interact with objects from a first-person view. To address this task, Huang et al. (2018) and Kazakos et al. (2021) focus on recognizing close-set EgoHOIs using additional multimodal cues (e.g., gaze, sound), while Wang et al. (2023a) adopt a self-supervised approach (He et al., 2022) to exploit visual information. Considering the abundant resources of third-person data, some works (Li et al., 2021b; Xu et al., 2023) aim to transfer view-agnostic knowledge from third-person videos to egocentric viewpoints. However, the unpredictable nature of open-world environments poses new challenges, requiring models to handle a variety of unseen concepts. Recent studies (Chatterjee et al., 2024; Wang et al., 2023b) seek to improve the understanding of open-vocabulary EgoHOI, but these efforts are either limited to specific domains like kitchens (Damen et al., 2021) or laboratories (Sener et al., 2022), or involving easy EgoHOI recognition that is well-solved by current egocentric models. A promising strategy to address these limitations involves egocentric video-language pretraining (Lin et al., 2022; Pramanick et al., 2023; Zhao et al., 2023; Zhang et al., 2023), which learns generalizable representations by leveraging the Ego4D (Grauman et al., 2022) dataset with over 3,000 hours of footage of daily human interactions. As a pioneering work, EgoVLP (Lin et al., 2022) uses the EgoNCE loss to treat video-text samples with similar HOIs as positives and visually similar videos as negatives during pretraining. Another method, LaViLa, enhances text supervision by generating diverse positive captions for videos to foster robust contrastive learning through a visual-conditioned GPT-2 (Radford et al., 2019) and a T5 (Raffel et al., 2020) rephraser. In this work, we expose the limitation of these EgoVLMs on recognizing HOI-related word variations and address the issue by improving the contrastive loss, which also benefits other downstream tasks.

**Hard Negative Mining.** Hard negative mining is a pivotal technique (Robinson et al., 2021; Zolfaghari et al., 2021) for refining representations within the visual-language metric space during contrastive learning. Traditionally, this process pairs positive samples with hard negatives that exhibit high feature similarity within pretraining datasets (Pramanick et al., 2023; Bao et al., 2022; Li et al., 2021a; Xu et al., 2021), or selecting hard negative from clips recorded in similar environment (Lin et al., 2022). However, despite current EgoVLMs have adopted either visual- or feature-based negative mining to enhance learning EgoHOI knowledge, they fall short in addressing the straightforward understanding task in EgoHOIBench. Recent innovations have introduced the generative negative sampling strategy using LLMs, aiming to enhance improve compositional understanding (Yuksekgonul et al., 2023) in image-VLMs (Radford et al., 2021; Li et al., 2022; Singh et al., 2022; Zeng et al., 2022), and action comprehension (Momeni et al., 2023; Bansal et al., 2024) in video-VLMs (Luo et al., 2022). For instance, ViA (Momeni et al., 2023) proposes a verb-focused pretraining framework that creates negative captions of sentences and verb phrases using an LLM. However, most existing approaches are tailored for pretraining on third-person datasets like MiT (Monfort et al., 2019) and applied to simple scenarios like Kinetics-400 (Carreira & Zisserman, 2017), leaving the complex egocentric domain underexplored. In this paper, our proposed learning objective, EgoNCE++, incorporates generative negative mining into the pretraining process, using either the powerful LLM or the more efficient vocabulary from the pretraining set to facilitate more robust EgoVLMs.

## 6 CONCLUSION

In this work, we introduce EgoHOIBench, a straightforward test designed to assess EgoVLMs' comprehension of HOI combinations, highlighting the current limitations of these models in understanding hand-object activities. We identify the underlying issues, including a lack of fine-grained negative text supervision and the object-centric feature space that favors HOI-noun recognition but adversely impacts HOI-verb recognition. Building upon these analyses, we propose an asymmetric learning objective called EgoNCE++, which enhances the video-to-text loss by incorporating generated dense hard negatives, and a text-to-video loss that focuses on grouping videos with similar nouns. Through extensive experimental analyses across diverse benchmarks, we demonstrate that our proposed VLP training framework can effectively equip different EgoVLMs with greater robustness to HOI combinations and benefit various downstream EgoHOI tasks.

ACKNOWLEDGEMENTS

This work was partially supported by the Beijing Natural Science Foundation (No. L233008) and the Outstanding Innovative Talents Cultivation Funded Programs 2024 of Renmin University of China.

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

## A  DISCUSSIONS

**Limitations.**  While EgoNCE++ delivers significant improvements across various fine-grained HOI benchmarks for multiple EgoVLMs, it does have some limitations. First, although hand-object interactions constitute a significant portion of egocentric activities, egocentric scenarios encompass a broader range of actions, such as simple observation of the environments or VR/AR activities like dancing or playing sports (Grauman et al., 2024b). A promising solution to this challenge is to incorporate third-person videos into the pretraining corpus (Dou et al., 2024) or leverage pretrained models based on third-person video data (Pei et al., 2024). Second, we find it challenging to enhance the EgoVLM's object recognition capabilities solely through text supervision. This difficulty likely stems from the broader diversity of object categories compared to action types in the real world, making effective capture challenging with a limited number of negative samples. Introducing visual supervision signals such as bounding boxes may be beneficial (Zhang et al., 2023; Phan et al., 2024). We plan to address these challenges in future works.

**Social Impact.**  The knowledge of EgoHOIs acquired by EgoVLMs holds great potential for real-world applications, including embodied agents and VR/AR systems. However, the use of egocentric videos raises privacy concerns, as they often capture personal and sensitive information. If not carefully managed, these privacy issues could lead to negative consequences. Furthermore, EgoVLMs are particularly relevant in contexts like kitchen environments, where recognizing dangerous activities is critical. Misinterpreting EgoHOIs could result in harmful outcomes, such as failing to recognize unsafe actions during tasks involving sharp objects. Our research addresses some of these challenges by demonstrating a more robust understanding of HOI actions, providing improved generalization and potentially mitigating these risks.

## B  MORE DETAILS OF THE EGOHOIBENCH

### B.1  CONSTRUCTION PROCESS

We develop EgoHOIBench based on EgoMCQ (Lin et al., 2022), which sources a diverse collection of 39,000 video clips from the validation set of the Ego4D dataset. In our curation process, we only keep those EgoHOI clips performed by the camera wearer, excluding clips that record other people's activities, such as multi-person interactions (Ryan et al., 2023). We achieve this by keeping the captions that begin with '#C' (denoting the wearer) and are followed by HOI-related verbs and nouns, while filtering out any notations related to other individuals, such as '#O'. To construct the

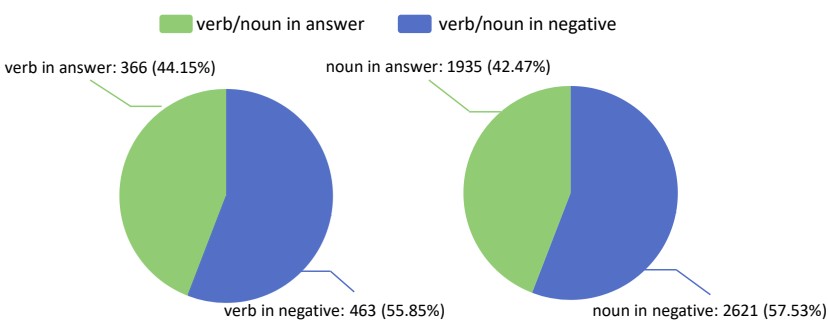

Figure 8: Illustration of the vocabulary statistics of EgoHOIBench.

HOI recognition trials as defined in our task definition above, given a video and its ground truth caption, we prompt an LLM to create candidate captions that contain semantically different HOIs from the ground truth. Specifically, we employ the LLaMA-3-8B (AI, 2024) model to generate HOI candidates through in-context learning. We provide the specific prompts and two exemplary tasks used in this process, along with examples of the final cases in Figure 14, which target generating words with different meanings to make the choices easier. To avoid semantic redundancies and ensure the uniqueness of the hard negative candidates, we use the Ego4D dictionary to eliminate possible synonyms from the generated captions. Ultimately, EgoHOIBench comprises 29,651 video clips, each accompanied by one ground truth caption, 10 negative captions with verb changes, and 10 negative captions with noun changes. Setting the number of negatives to 10 (i.e. 10 noun negatives, 10 verb negatives) forms 100 HOI negatives, which aligns with the typical action recognition setting.

## B.2 VOCABULARY STATISTICS

The statistics of the vocabulary information are presented in Figure 8. This dataset features a rich and diverse vocabulary, including approximately 800 verbs and 4,000 nouns. The options generated by LLMs effectively double the vocabulary size compared to the original correct answers, resulting in extensive combinations of verbs and nouns.

## C MORE EXPERIMENTAL ANALYSIS

For fair comparisons, we have re-implemented all experiments in the same environment and under identical settings, without any adjustments to the hyperparameters.

### C.1 IMPLEMENTATION DETAILS

As introduced in the main paper, we validate our approach on three EgoVLMs including EgoVLP, its advanced version EgoVLPv2, and LaViLa. EgoVLP is pretrained on the EgoCLIP-3.8M dataset and employs the EgoNCE loss for optimization. EgoVLPv2 enhances the original model by incorporating a cross-attention mechanism between dual encoders and by pretraining on additional proxy tasks. LaViLa, on the other hand, is trained on a vast dataset of 4 million videos, with 56 million captions generated by a visual-conditioned GPT-2 (Radford et al., 2019) and further refined using a sentence rephraser T5 (Raffel et al., 2020). This extensive training regimen enables LaViLa to improve the generalization of EgoVLMs. We present a summary of well-known EgoVLP methods in Table 5. Our proposed EgoNCE++ further enhances the EgoHOI understanding capabilities of pretrained EgoVLM models, utilizing only few trainable parameters and a novel pretraining objective.

For all models, we adopt the AdamW optimizer with parameters $\beta_1 = 0.9$ and $\beta_2 = 0.999$. The learning rate follows a cosine annealing schedule, starting at 3e-5 and gradually reducing to 3e-7. During training, we apply standard RandomResizedCrop for data augmentation and employ LoRA tuning to continuously pretrain our EgoVLM. The text encoder for EgoVLP is DistilBERT (Sanh et al., 2019), while LaViLa uses CLIP (Radford et al., 2021). In the case of EgoVLPv2, we implement a dual encoder architecture without cross-attention fusion, training it exclusively with EgoNCE++. The text encoder for EgoVLPv2 is RoBERTa (Liu et al., 2019). It is important to note that in the

Table 5: Summary of existing egocentric video-language pretraining methods compared with ours.

| METHOD | Pretrain Data | Objective | Negative Mining | Visual Encoder | Text Encoder | Train Param |
|--------|---------------|-----------|-----------------|----------------|--------------|-------------|
| EgoVLP | 3.8M | EgoNCE | video sim | ImageNet | DistillBert | 172M |
| EgoVLPv2 | 3.8M | EgoNCE+MLM+VTM | feature sim | ImageNet | Roberta | 364M |
| LaViLa | 4M | InfoNCE | none | CLIP | CLIP | 180M |
| Ours | 2.5M | EgoNCE++ | text sim | EgoVLM | EgoVLM | +3M-43M |

text encoder, for LaViLa, we fine-tune the word embeddings, while for the other two EgoVLMs, the entire text encoder remains frozen. Our experiments show that fine-tuning the word embeddings in EgoVLP and EgoVLPv2 results in reduced generalization performance across public benchmarks such as EK-100-MIR. The difference may result from the tokenizers, where only LaViLa uses the BPE tokenizer (Sennrich et al., 2016).

We implement all models using their original codebases, with one exception: videos are loaded using the Decord library, as recommended by LaViLa, instead of using pre-extracted frames via FFmpeg. This may lead to slight numerical differences in the results for pretrained EgoVLP and EgoVLPv2 compared to the figures reported in their original papers.

## C.2 BENCHMARK DETAILS

**Multi-Instance Retrieval in EK-100-MIR.** For the zero-shot setting, we conduct video-text matching for retrieval tasks, using 16 frames for evaluation. For the fine-tune setting, we finetune the EgoVLMs using the AdamW optimizer. The learning rate is dynamically adjusted from 3e-3 to 1e-5 using a cosine annealing scheduler that incorporates a linear warmup, starting at 1e-6 for the first epoch. We deploy a total batch size of 128 across 8 GPUs. During both training and inference, 16 frames are sampled from each video.

**Action Recognition in EGTEA.** For the zero-shot setup, we evaluate mean results across all evaluation splits, as suggested by Li et al. (2018), by conducting a video-text retrieval task between video clips and their corresponding action text labels. We prepend the text labels with the prompt "#C C ..." to standardize the input format. For the fine-tuning setup, we leverage the visual encoder and attach an additional linear projection head for the classification purpose, following Kazakos et al. (2021). The models are trained and evaluated on the first split of the validation set. We employ the same optimizer, scheduler, batch size, and frame sampling rate as used in EK-100-MIR. At inference time, we perform three spatial crops of size $224 \times 224$ from each $256 \times 256$ frame of the video clip, averaging their predictions to form the final prediction.

**Action Recognition in CharadesEgo.** We treat action recognition as a video-text retrieval task, where video clips are matched with their corresponding action text labels in a zero-shot evaluation setting. During inference, we sample 16 frames from each video. Notably, previous works evaluate their models on CharadesEgo using the initial checkpoint due to the domain gap problem, where continued training often leads to performance drops (Lin et al., 2022). In contrast, these studies use their best checkpoint for evaluation on other datasets, such as EK100. In our experiments, EgoNCE++ continues pretraining from the best checkpoints instead of starting from the initial checkpoint of EgoVLMs, to ensure a fair comparison and consistent pretraining setting between EgoVLM and EgoVLM++. Consequently, it is common to observe lower numbers in our paper than those reported in their original papers. Given the consistent improvements offered by EgoNCE++, we believe that if we were to pretrain from their initial checkpoints, EgoVLMs would still benefit from EgoNCE++.

**Action Recognition in EK100-CLS.** For the zero-shot setting, we organize the task similarly to EgoHOIBench. For verb classification, we append the ground truth noun, while for noun classification, we prepend the ground truth verb. In the linear probing setting, we freeze the visual encoder and add a linear layer to map the feature embeddings to the predefined classes.

Table 6: Comparison of open-vocabulary action recognition on the EK-100-OV dataset.

| METHOD | HOI DETECTOR | TYPE | OPEN-SET | | CLOSE-SET | |
| | | | top-1 action (%) | top-5 action (%) | top-1 action (%) | top-5 action (%) |
|---|---|---|---|---|---|---|
| S3D | ✓ | fine-tune | 0 | - | 37.6 | - |
| 2×S3D | ✓ | fine-tune | 0.1 | - | 36.7 | - |
| OAP+AOP | ✓ | fine-tune | 11.2 | - | 35.9 | - |
| LaViLa | ✗ | zero-shot | 7.57 | **22.78** | 16.59 | 34.88 |
| LaViLa++ | ✗ | zero-shot | **8.48** | 21.36 | **17.34** | **36.96** |

Table 7: Comparison on downstream benchmarks under the zero-shot setup, where "MODEL++" denotes using EgoNCE++ to continue to pretrain the original MODEL.

| METHOD | EgoHOIBench | | | Epic-Kitchens-100-MIR | | | | | | CharadesEgo |
| | verb (%) | noun (%) | action (%) | mAP (%) | | | nDCG (%) | | | mAP |
| | | | | V→T | T→V | Avg. | V→T | T→V | Avg. | |
|---|---|---|---|---|---|---|---|---|---|---|
| EgoVLP | 40.27 | 68.60 | 30.16 | 25.2 | 19.2 | 22.2 | 28.1 | 25.4 | 26.7 | 19.3 |
| EgoVLP++ | **56.11** | **69.05** | **41.63** | **25.6** | **19.7** | **22.7** | **28.6** | **25.7** | **27.1** | **19.7** |
| EgoVLPv2 | 36.10 | 63.40 | 26.40 | 26.9 | 19.9 | 23.4 | 28.8 | 26.8 | 27.8 | 17.2 |
| EgoVLPv2++ | **44.41** | **64.10** | **32.40** | **28.0** | 19.9 | **23.9** | **29.8** | 26.8 | **28.3** | **17.5** |
| LaViLa | 46.61 | 74.33 | 36.85 | 35.1 | 26.6 | 30.8 | 33.7 | 30.4 | 32.0 | 20.6 |
| LaViLa++ | **80.63** | **75.30** | **63.17** | **35.8** | **27.5** | **31.7** | **33.9** | **30.7** | **32.3** | **20.9** |

## C.3 MAIN RESULTS

### C.3.1 ZERO-SHOT SETUP EVALUATION

**EgoNCE++ Consistently Improves Generalization Across All EgoVLMs.** As shown in Table 7, all EgoVLMs benefit from pretraining with EgoNCE++, resulting in consistent improvements across EgoHOIBench, EK100-MIR and CharadesEgo.

**Open-Set EgoHOI Recognition on EK-100-OV.** The EK-100-OV (Chatterjee et al., 2024) aims to recognize unseen categories, especially novel objects, at inference time. We evaluate both LaViLa and LaViLa++ on this benchmark in the zero-shot setup, with results presented in Table 6. Although our model does not outperform those specifically designed models that extract object region features using an HOI detector (Shan et al., 2020) at inference time, it demonstrates strong generalization capabilities and competitive results on top-5 actions, consid-

Table 8: Comparison of action recognition tasks on Epic-Kitchens-100, where the zero-shot setting is organized the same way as EgoHOIBench.

| SETTING | METHOD | verb | noun | action |
|---|---|---|---|---|
| Zero-Shot | LaViLa | 11.65 | 39.78 | 10.14 |
| | LaViLa++ | **16.10** | **43.35** | **14.67** |
| Linear Probing | LaViLa | 59.06 | 41.53 | 26.29 |
| | LaViLa++ | **59.43** | **42.41** | **26.86** |

ering 2,639 candidate HOI combinations at inference time. Compared to LaViLa, our enhanced model LaViLa++ shows clear improvement across most key metrics (e.g., **+0.91%** in open-set top-1 action accuracy), highlighting its effectiveness in adapting to open-set conditions.

**LaViLa++ Exhibits Better Linear Probing Property.** We conduct zero-shot classification and linear probing on EK100-CLS, as shown in Table 8, which highlights the improvement in the video feature space and generalization capabilities of LaViLa++. We achieve a steady improvement of **+4.53%** on action accuracy under zero-shot settings and **+0.57%** under linear probing settings.

**Larger Model Sizes Still Benefit from EgoNCE++.** EgoNCE++ can also be applied to the LaViLa-Large model, as shown in Table 9. First, LaViLa++ enhances the model's robustness to HOI-related word variations, demonstrated by a significant improvement of **+27.01%** on EgoHOIBench. Additionally, our model achieves notable gains of **+1.0%** in mAP and **+1.1%** in nDCG on other datasets. While our model surpasses the HelpingHands model (without multitask from the HOI detection) by an average of +0.8% in mAP, its overall performance still falls short. The HelpingHands model freezes LaViLa-Large and adds a transformer decoder to learn an object-aware feature space through multitask learning, combining enhanced video-language pretraining with video-noun match-

Table 9: Comparison of models built upon LaViLa-Large on zero-shot EK-100-MIR and EGTEA.

| METHOD | EgoHOI-B | Epic-Kitchens-100-MIR | | | | | | EGTEA | |
| | action acc | mAP (%) | | | nDCG (%) | | | mean-acc | top1-acc |
| | | V→T | T→V | Avg. | V→T | T→V | Avg. | | |
| LaViLa | 38.69 | 40.0 | 32.2 | 36.1 | 36.1 | 33.2 | 34.6 | 34.1 | 40.1 |
| LaViLa++ | **65.70** | **41.3** | **32.8** | **37.1** | 37.8 | 33.6 | 35.7 | 37.5 | 38.6 |
| HelpingHands | - | 42.3 | 32.7 | 37.5 | 39.3 | 36.2 | 37.8 | 39.1 | 46.6 |
| HelpingHands w/o obj | - | 40.7 | 31.1 | 35.9 | **38.3** | **35.0** | **36.6** | **44.9** | **40.1** |

Table 10: Comparison with state-of-the-arts on EK-100-MIR and EGTEA under the fine-tune setup.

| METHOD | Epic-Kitchens-100-MIR | | EGTEA | |
| | mAP (%) | nDCG (%) | top-1 acc | mean acc |
| MME (Wray et al., 2019) | 38.5 | 48.5 | - | - |
| JPoSE (Wray et al., 2019) | 44.0 | 53.5 | - | - |
| LSTA (Sudhakaran et al., 2019) | - | - | 61.86 | 53.00 |
| IPL (Wang et al., 2021) | - | - | - | 60.15 |
| MTCN (Kazakos et al., 2021) | - | - | 73.59 | 65.87 |
| LaViLa (Zhao et al., 2023) | **50.4** | 64.8 | 78.04 | 70.56 |
| LaViLa++ | 50.1 | **65.1** | **78.33** | **71.20** |

ing, video-sentence matching via EgoNCE, and HOI detection using pseudo-labels. Although the HelpingHands model is generally stronger, the frozen features from LaViLa would still struggle on EgoHOIBench. We believe our method is orthogonal and compatible with models like HelpingHands. By replacing EgoNCE with EgoNCE++, we propose that these objectives could collaboratively strengthen EgoVLMs by providing both better visual supervision and text supervision.

### C.3.2 FINE-TUNING SETUP EVALUATION

In this setup, we further finetune the model on the training and validation splits of downstream tasks.

**Multi-Instance Retrieval on EK-100-MIR.** As illustrated in Table 10, LaViLa++ outperforms its original version in terms of nDCG but shows a decrease in mAP. These results suggest that while our approach enhances the ranking of candidates, it does not retrieve data with similar HOIs as effectively, highlighting a trade-off between fine-grained HOI recognition and the diversity of retrieved outcomes. Despite this, LaViLa++ still serves as a strong zero-shot learner for EgoHOI actions.

**Action Recognition on EGTEA.** This benchmark specifically focuses on cooking activities. Notably, LaViLa++ achieves state-of-the-art performance on EGTEA, showcasing its ability to leverage the robust generalization capabilities of LaViLa. The improvement observed on EGTEA demonstrates that our proposed approach remains effective even when evaluated on out-of-domain benchmarks.

### C.4 MORE ABLATION STUDIES

Ablation studies are conducted by pretraining the EgoVLP model (Lin et al., 2022) using EgoNCE++.

**Type of Negatives in V2T.** The impact of verb ("VERB") or noun ("NOUN") negatives generated by the LLM is detailed in Table 11. Negative verb samples effectively enhance model training, improving verb accuracy by **+14.89%**. In contrast, noun negatives yield a modest impact with an accuracy improvement of +0.17%. This discrepancy could be attributed to the noun vocabulary size of approximately 7k words, which is considerably larger than the verb vocabulary of about 2k words, making it more challenging to acquire visual knowledge from text supervision with limited data.

Table 13: Ablation of different training strategies.

| VIS | TEXT | PARAM | EgoHOIBench | EK-100-MIR | |
| | | | action acc | mAP | nDCG |
| frozen | frozen | 0M | 30.16 | 22.2 | 26.7 |
| LoRA | frozen | 3.1M | 41.63 | 22.7 | 27.1 |
| full | frozen | 109M | 44.39 | 22.4 | 27.0 |
| frozen | full | 63.5M | 60.18 | 9.6 | 16.8 |
| LoRA | full | 66.7M | 60.01 | 9.8 | 16.9 |
| full | full | 172.5M | 59.82 | 12.5 | 19.2 |

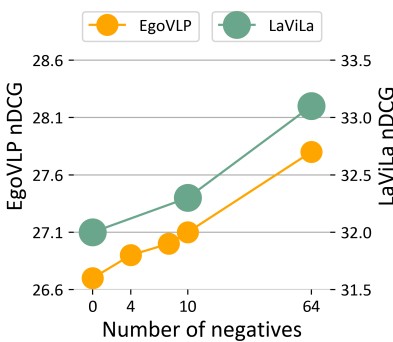

Figure 9: Scaling effect of negative number on EK-100-MIR (avg. nDCG).

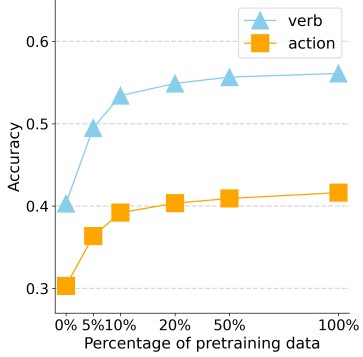

Figure 10: Impact of varying data volume used in pretraining.

Table 11: Ablation of types of negatives.

| VERB | NOUN | verb (%) | noun (%) | action (%) |
|---|---|---|---|---|
| ✗ | ✗ | 40.70 | 68.86 | 30.51 |
| ✗ | ✓ | 41.47 | 69.06 | 31.29 |
| ✓ | ✗ | 55.16 | 69.03 | 40.81 |
| ✓ | ✓ | 55.29 | 69.03 | 40.88 |

Table 12: Ablation of types of positives.

| VERB | NOUN | verb (%) | noun (%) | action (%) |
|---|---|---|---|---|
| ✗ | ✗ | 55.29 | 69.03 | 40.88 |
| ✓ | ✓ | 55.34 | 68.89 | 41.00 |
| ✓ | ✗ | 55.93 | 68.80 | 41.26 |
| ✗ | ✓ | 56.11 | 69.05 | 41.63 |

**Type of Positives in T2V.** Table 12 shows different positive sampling strategies in T2V loss. These strategies aggregate video representations based on verbs (VERB) or nouns (NOUN) in their captions. Due to the strong bias towards nouns, the results show that aggregating nouns alone yields the largest improvements, whereas pulling videos with similar verbs slightly damages noun recognition.

**Training Strategy for Dual Encoder.** We further investigate the impact of various training strategies for dual encoders, as shown in Table 13. Comparing row 2 and row 3, we observe that full tuning outperforms LoRA tuning by +2.76% on EgoHOIBench but underperforms by an average of -0.3% on EK-100-MIR. These results indicate that while using additional parameters during full tuning can improve performance, it may also lead to decreased generalization on out-of-domain benchmarks. Given the importance of generalization in the real world, we opt for LoRA tuning for the visual encoder while keeping the text encoder frozen. When the text encoder is trainable, as shown in rows 4-6, there is a boost in performance on EgoHOIBench, even approaching the results achieved by LaViLa++. However, the lack of generalization to EK-100-MIR suggests significant overfitting to the pretraining dataset. Therefore, we choose to freeze the text encoder to ensure generalization.

**Volume of Used Pretraining Data.** Results on the pretraining data size are presented in Figure 10. The findings highlight a significant increase when only 10% of the data (250K) is used, with action accuracy rising from 30.3% to 39.2%. In contrast, using the remaining data only results in an improvement of +2.43%.

**LoRA Rank.** We conduct another study to investigate the impact of rank configurations for LoRA. As detailed in Table 14, our findings reveal that a LoRA rank of 16 enhances generalization capabilities, while even a minimal rank of 1 can significantly improve EgoHOI recognition performance. This trend suggests that relatively small training adjustments can significantly enhance the visual feature space, leading to improved performance with minimal computational cost.

Table 14: Ablation of LoRA rank.

| LoRA | PARAMS | verb (%) | noun (%) | action (%) |
|---|---|---|---|---|
| 1 | 0.24M | 54.76 | 68.86 | 40.52 |
| 4 | 0.82M | 55.11 | 68.89 | 40.95 |
| 16 | 3.14M | 55.29 | 69.03 | 40.89 |
| 32 | 6.24M | 55.01 | 68.80 | 40.64 |

**Scaling Effect of Negative Numbers on nDCG.** Figure 9 illustrates the trend in nDCG for EK-100-MIR as the number of negative samples increases. Similar to mAP, there is a clear correlation where using more negatives leads to better performance. While mAP focuses on identifying the single

Table 15: Comparison of results on EgoHOIBench generated from different LLMs, where "MODEL++" denotes using EgoNCE++ to continue to pretrain the original MODEL.

| METHOD | LLaMA-EgoHOIBench | | | GPT4o-EgoHOIBench | | | DeepSeek-EgoHOIBench | | |
|---|---|---|---|---|---|---|---|---|---|
| | verb (%) | noun (%) | action (%) | verb (%) | noun (%) | action (%) | verb (%) | noun (%) | action (%) |
| EgoVLP | 40.27 | 68.60 | 30.16 | 41.44 | 66.55 | 33.96 | 40.75 | 56.42 | 30.56 |
| EgoVLP++ | 56.11 | 69.05 | 41.63 | 50.89 | 70.19 | 41.78 | 50.59 | 60.84 | 38.04 |
| LaViLa | 46.61 | 74.33 | 36.85 | 45.18 | 72.82 | 38.20 | 44.16 | 62.28 | 34.09 |
| LaViLa++ | 80.63 | 75.30 | 63.17 | 53.95 | 73.65 | 44.63 | 52.73 | 63.48 | 39.15 |

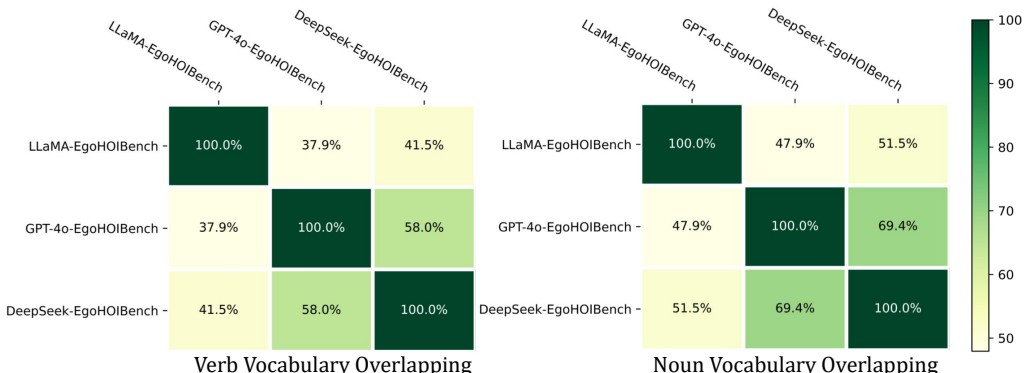

Figure 11: Vocabulary overlap ratio of EgoHOIBench generated by different LLMs. Word frequencies in the vocabulary are normalized, resulting in a symmetric matrix.

Table 16: Analysis of the efficiency and effectiveness of different negative generators. The vocabulary-based method offers efficient online generation, while the LLM-based approach provides greater generalization at a higher computational cost. The results are derived from EgoVLP++.

| METHOD | SPEED (ms/neg) | EgoHOIBench | | | EK-100-MIR | |
|---|---|---|---|---|---|---|
| | | verb (%) | noun (%) | action (%) | mAP | nDCG |
| none | - | 40.27 | 68.60 | 30.16 | 22.2 | 26.7 |
| vocab-based | 0.005616 (on CPU) | 54.46 | 68.56 | 40.07 | 22.5 | 27.1 |
| LLM-based | 27.648 (on GPU) | 56.11 | 69.05 | 41.63 | 22.7 | 27.1 |

correct answer, nDCG emphasizes the overall quality of the ranking. The improved nDCG indicates that increasing the number of negatives helps refine the ranking, elevating more relevant HOIs and relegating irrelevant ones, thereby enhancing HOI understanding.

## C.5 MORE ANALYSES

**Bias of LLMs in EgoHOIBench.** To reveal bias on the negative generation in LLMs, we create Ego-HOIBench similarly using other LLMs: DeepSeek-200B (DeepSeek-AI, 2024) and GPT-4o (OpenAI, 2024). In Figure 11, we calculate the vocabulary overlapping ratio, where the value in position $(i, j)$ represents the proportion of the vocabulary from the $i$-th set that overlaps with the $j$-th set. The GPT-4o and DeepSeek tend to generate different words compared with LLaMA. The evaluation results are shown in Table 15. The vocabulary bias affects the results to some extent, but our EgoNCE++ steadily improves the performance on all benchmarks.

**Efficiency Analyses of Vocab-Based and LLM-Based Negative Generators.** We analyze the trade-off between the efficiency and efficacy of our designed vocab-based and LLM-based negative generator in Table 16. Our LLM-based approach, which uses LLaMA-3-8B to generate HOI candidates, operates offline and requires a higher computational cost. Specifically, generating training data for 2.5M samples required approximately 192 GPU hours on A6000 hardware. In comparison, a vocab-based negative sampling method operates far more efficiently, requiring only 0.039 hours on 16 CPU threads and enabling online updates. Since both methods demonstrate notable improvements

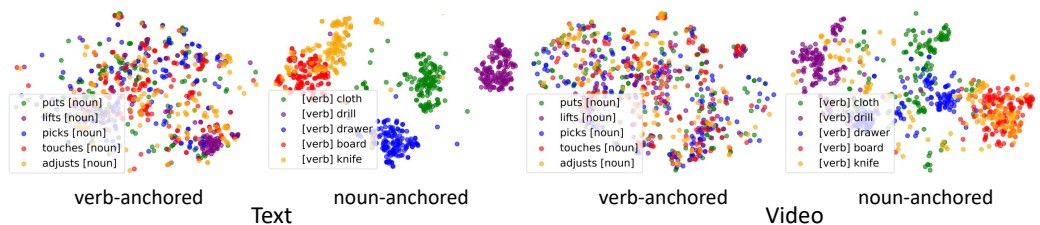

Figure 12: Visualization of EgoVLP's feature space. Both video and text feature space keep exhibiting an object-centric feature space.

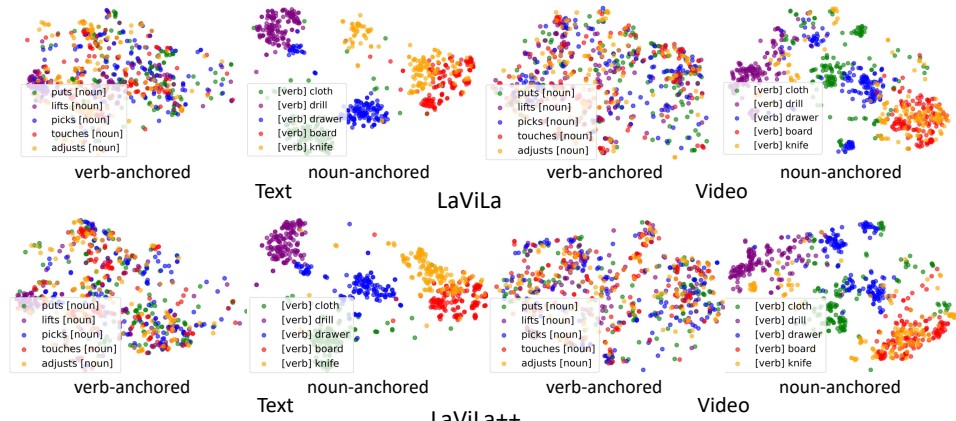

Figure 13: Comparison of LaViLa and LaViLa++'s feature spaces. Although they keep exhibiting object-centric feature spaces, the video-text matching is greatly improved by EgoNCE++.

on EgoHOIBench and EK-100-MIR, we recommend selecting the appropriate method based on the size of the pretraining vocabulary and the available computational resources.

## D QUALITATIVE RESULTS

**EgoVLP's Also Exhibits Object-Centric Feature Space.** Since EgoVLPv2 used in our paper (i.e., dual encoder without fusion in the backbone) shares a similar structure with EgoVLP, we visualize the EgoVLP's feature space in the same way as LaViLa, which also exhibits an object-centric characteristic. We suspect that most of the egocentric video-language models pretrained with contrastive learning will be object-centric, regardless of their detailed architecture. To illustrate why representations tend to cluster by nouns when verbs vary, we consider both the pretraining data and the video encoding architecture. From a data perspective, for example, a video of someone "cutting grass" is more visually similar to one of "watering grass" in the same environment (Grauman et al., 2022), whereas "cutting onion" in a kitchen would appear quite different from "cutting grass" due to the change in both verb semantics and visual content. From an architectural perspective, current vision models primarily encode videos based on single-frame visual information (Lei et al., 2023), focusing on objects rather than actions. As a result, the model tends to group representations by nouns (visual similarity) rather than verbs (temporal information). To create a more verb-friendly feature space, a potential solution could be incorporating multi-modality data that captures motion and temporal dynamics, such as optical flow or event cameras.

**Negatives Sampled from Different Generators.** We provide several examples of negative samples produced by different generators in Figure 15. Notably, LLM-based captions tend to be more semantically plausible than those generated by vocab-based or rule-based methods, which may include words not found in the Ego4D dictionary.

**Comparison before/after Using EgoNCE++.** As previously discussed, EgoVLP++ significantly outperforms EgoVLP after pretraining with EgoNCE++. To illustrate this, we provide examples of both improved cases and bad cases in Figure 16 and in Figure 17, respectively. Figure 16 shows that EgoNCE++ enhances the model's ability to learn more robust video-text alignments, enabling our refined model to identify fine-grained EgoHOIs. In contrast, Figure 17 highlights some extreme cases where our model struggles. In these cases, the background tends to be more complex, and the differences among actions are subtle, making them difficult to differentiate. We also visualize the feature space of LaViLa and LaViLa++ in Figure 13. These visualizations reveal that the feature space preserves its object-centric nature. However, the inherent structure of the video or text representation space does not necessarily impact the video-text matching results Instead, our approach enhances video-text matching by making the correct video-text pair more distinguishable from negative video-text ones.

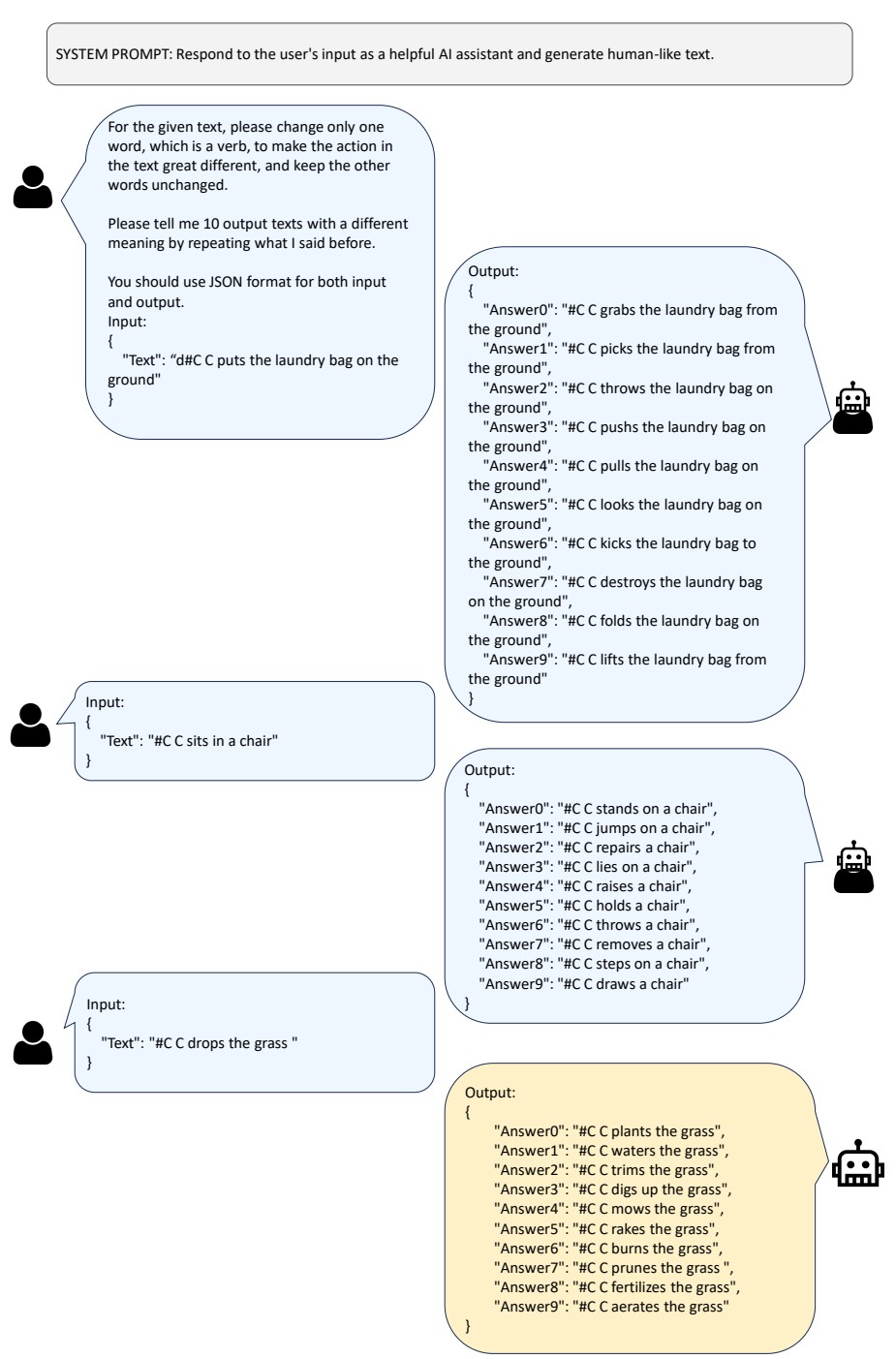

Figure 14: Examples of options on EgoHOIBench generated by LLM's in-context learning.

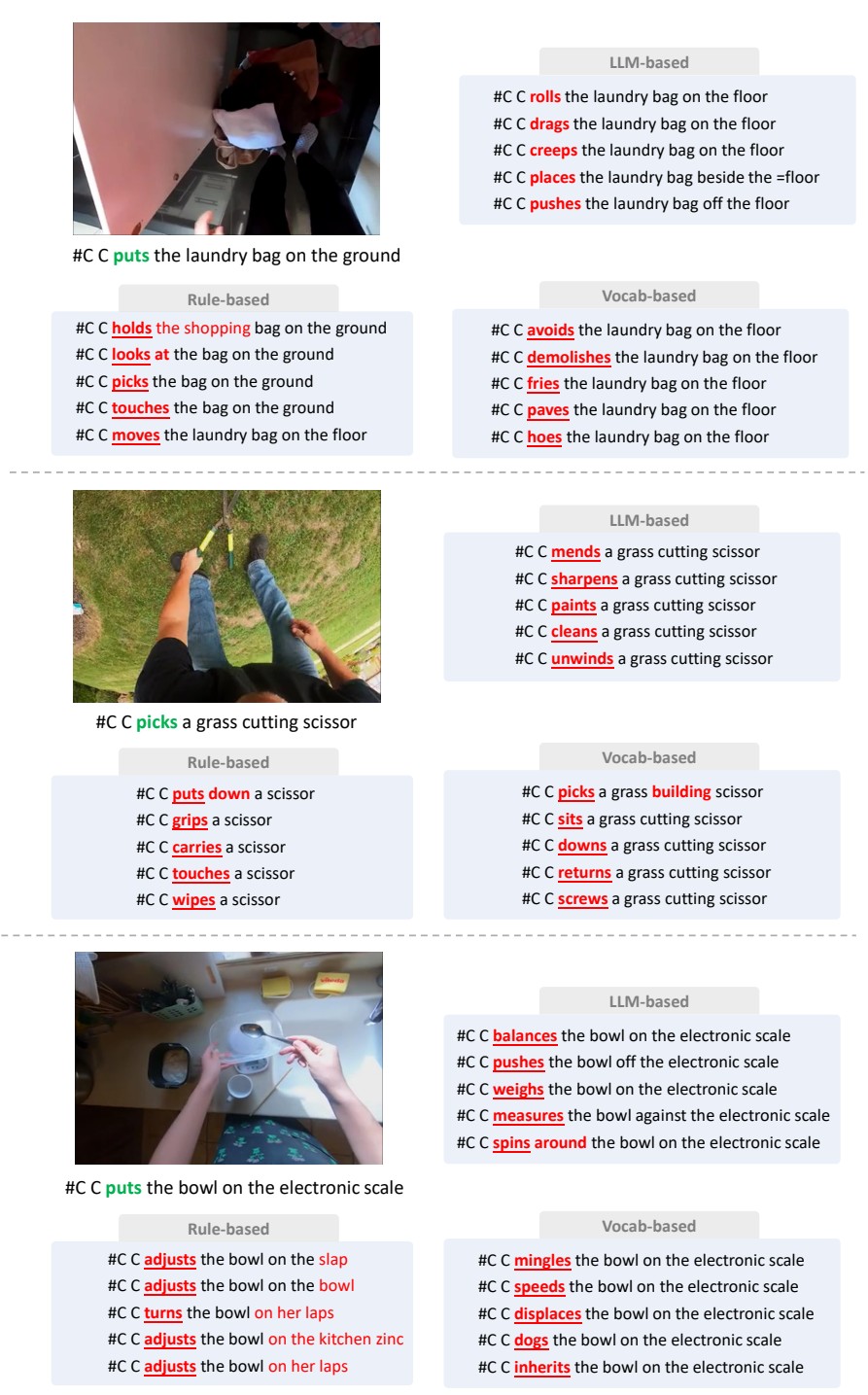

Figure 15: Examples of options generated by LLM in the pretraining set. We provide five candidates for simplicity. The green words denote the word to be replaced while the red ones denote words generated by different strategies.

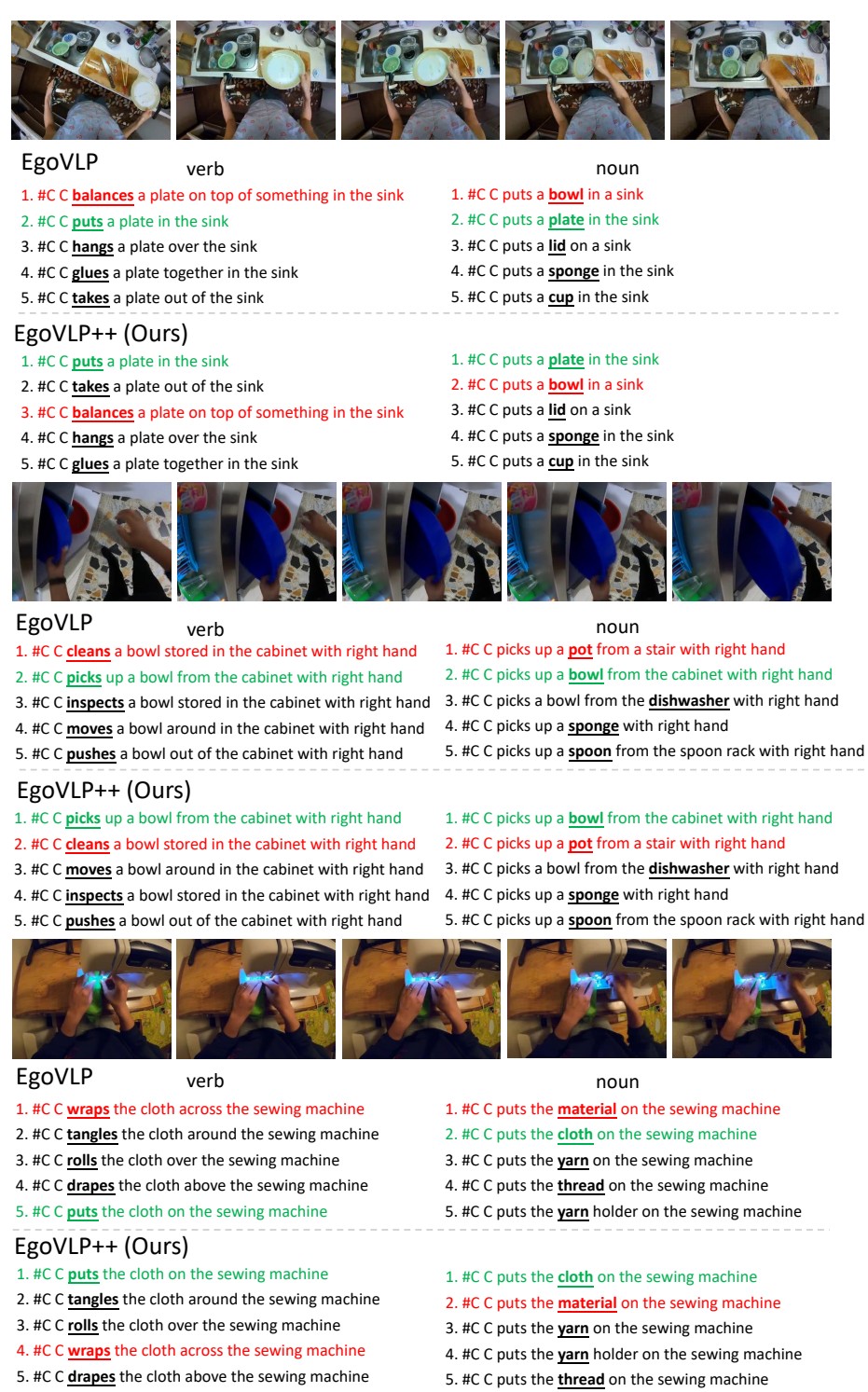

Figure 16: Improved cases on EgoHOIBench after using EgoNCE++. Five candidates are provided for simplicity. The green sentences denote ground-truth caption that is correctly classified by EgoVLP++ while the red ones are false positives predicted by EgoVLP.

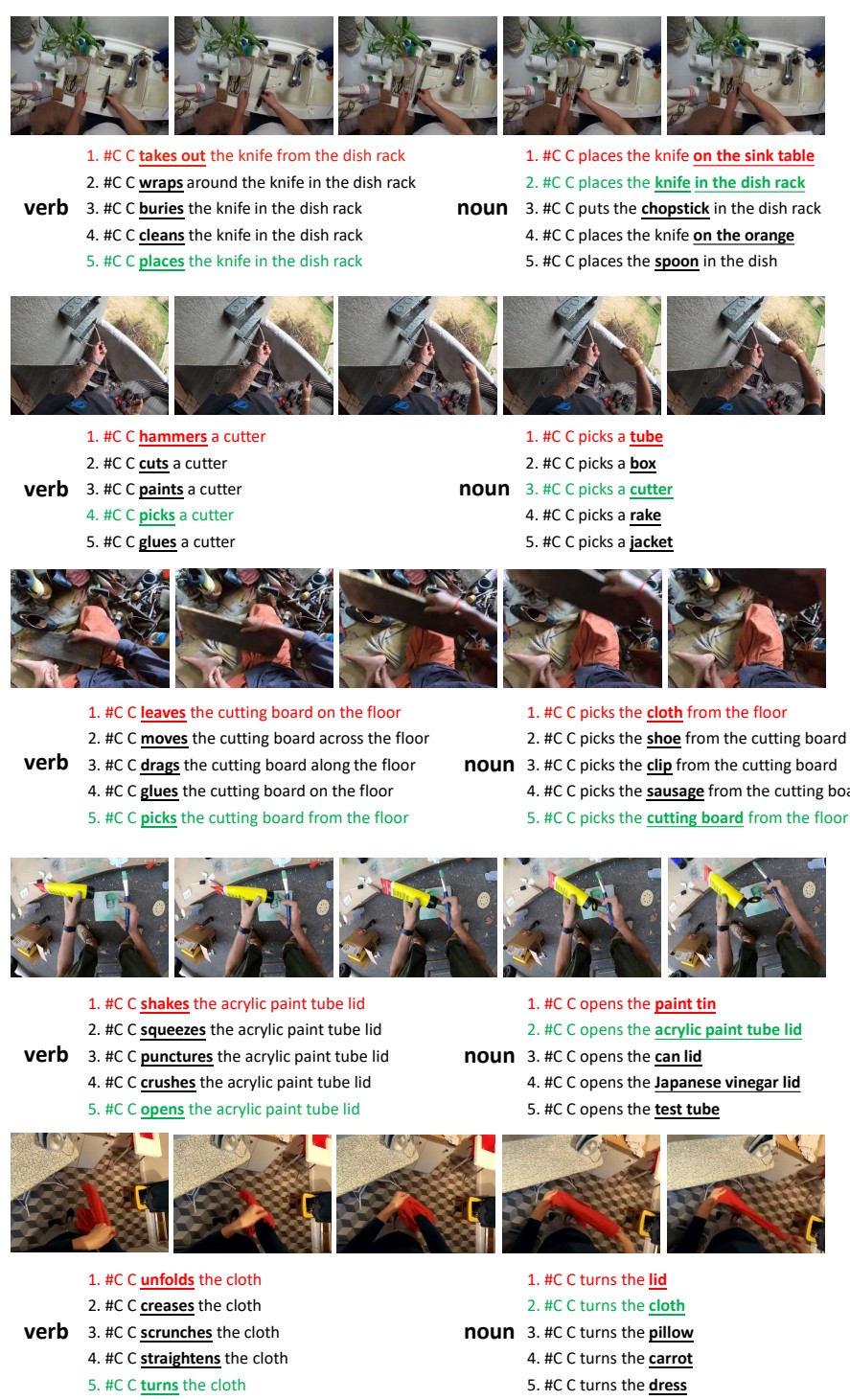

Figure 17: Bad cases on EgoHOIBench where EgoVLP++ struggles. The green sentences are ground truth and the red ones are mistakenly predicted by EgoVLP++. Others are the remaining candidates. Five candidates are provided for simplicity.

