# OpenReview forum: "Do Egocentric Video-Language Models Truly Understand Hand-Object Interactions?"
_ICLR.cc/2025/Conference — ICLR 2025 Poster_

### Official Review · Reviewer_3po1 · 2024-11-01

**Soundness:** 3
**Presentation:** 3
**Contribution:** 2
**Rating:** 6
**Confidence:** 4

**Summary:**

* Current EgoVLMs, used to understand hand-object interactions in first-person views, can be easily misled by minor changes in interaction descriptions (e.g., changing verbs or nouns).
* A benchmark was created to highlight the limitations of EgoVLMs, particularly their struggle with recognizing verbs compared to nouns, often due to insufficient fine-grained supervision.
* An asymmetric contrastive objective was introduced to improve video-language alignment:
 Video-to-Text Objective utilizes enhanced text supervision with negative captions generated by large language models or HOI-related vocabulary substitutions.
Text-to-Video Objective focuses on an object-centric feature space that clusters video representations based on shared nouns.
* The proposed approach, EgoNCE++, significantly improves EgoVLM performance, especially in tasks like multi-instance retrieval, action recognition, and temporal understanding.

**Strengths:**

1. The study tackles a key shortcoming in egocentric video-language models (EgoVLMs), which is their limited ability to distinguish subtle changes in interaction descriptions. This issue has significant implications for understanding hand-object interactions, a critical area in ego-centric vision applications.

2. The work introducing a specialized benchmark to evaluate EgoVLMs under challenging scenarios is a valuable contribution. This benchmark exposes performance gaps that were previously under-explored, providing a robust foundation for future research and model improvements.

**Weaknesses:**

1. The proposed asymmetric contrastive objective may lack the degree of methodological novelty expected. Similar objectives and contrastive learning techniques have been explored in other contexts as follows:
 * (a) The use of negative mining in contrastive learning is well explored in [1], where it aims to find better hard negatives to benefit contrastive learning.
 * (b) In the field of using augmented captions in multimodal contrastive learning is explored in [2].
 * (c) The use of LLM to generate more text for learning is explored in [3].


2. Since the model is trained and evaluated on a benchmark specifically designed for the study, it’s unclear if these gains will translate to other existing benchmarks or real-world applications. As shown in Tables 3 and 4, the gain over the existing dataset using the proposed dataset is limited. This creates the risk of overfitting to a controlled benchmark, which could raise questions about generalizability.

[1] Robinson, Joshua, et al. "Contrastive learning with hard negative samples." ICLR 2021.

[2] Yuan, Xin, et al. "Multimodal contrastive training for visual representation learning." Proceedings of the IEEE/CVF Conference on Computer Vision and Pattern Recognition. 2021.

[3] Fan, Lijie, et al. "Improving clip training with language rewrites." Advances in Neural Information Processing Systems 36 (2024).

**Questions:**

1. How does the proposed asymmetric contrastive objective compare with other contrastive techniques in video-language modeling? Could you clarify what specific elements make it novel in this context?

2. Can you elaborate on the robustness of the generated negative captions? Have you evaluated how biases in large language models might affect the model's performance?

3. How does the proposed benchmark compare to existing ones since the improvement is not significant on the existing dataset, and do you anticipate any limitations when applying it to real-world scenarios or other egocentric datasets?

---

> ### Author Response · Authors · 2024-11-22
> **Official Comment to Reviewer 3po1‘s Weaknesses [1/2]**
>
> Thank you for your detailed review and constructive feedback on our paper. We have carefully considered your comments and below we provide our responses to address your concerns. Note that we refer to the table/figure numbers in our revised paper.
>
> **W1: The Asymmetric Contrastive Objective Lacks the Degree of Methodological Novelty Expected**
>
> We provide comparison with previous works: Both the asymmetric design of EgoNCE++ and the LLM-based / vocab-based hard negative mining in V2T loss represent novel contributions to egocentric video understanding.
> - **In Egocentric Video-Language Modeling**: Unlike prior pretraining symmetric objectives, our asymmetric EgoNCE++ considers from both vision-to-text (V2T) negative mining and text-to-vision (T2V) positive sampling perspectives within a unified contrastive loss framework. This design effectively addresses the unique challenges of egocentric video data. As shown in our results on EgoHOIBench, existing hard negative mining techniques struggle to make these fine-grained distinctions (negative mining techniques are listed in Table 5 in the supplementary material, where EgoVLP adopts negative mining based on visual similarities and EgoVLPv2 mines negatives that have similar features), while our method consistently improves robustness and generalization across multiple datasets. Our method also surpasses state-of-the-art works as illustrated in Table 1.
> - **In Hard Negative Mining**: Our use of vocab-based augmentation for efficiently generating tailored negatives offers a new technical insight. This approach significantly boosts performance without relying on large-scale language models, which is a limitation of LaCLIP [3]. Specifically, generating training data for 2.5M samples took approximately 192 GPU hours on A6000 hardware (i.e., 0.05616 ms per text sample to generate 10 negatives). In contrast, the vocab-based negative sampling method  requires only 0.039 hours on 16 CPU threads (i.e., 0.05616 ms per text sample to generate 10 negatives), enabling efficient online updates. This efficiency and effectiveness are critical for advancing understanding in egocentric contexts.
>
> **W2: The Risk of Overfitting to a Controlled Benchmark and The Generalizability of Our Model.**
>
> We understand the concern about the generalizability of our model. While our benchmark is tailored to evaluate fine-grained hand-object interactions, we emphasize that its data sources—the largest egocentric dataset Ego4D—span diverse scenarios and represent real-world variability. This ensures that our pretrained models exhibit robust performance across related tasks.
> - **Performance on Other Benchmarks**: While our ablation studies in Table 3 and Table 4 were primarily conducted on EgoVLP, our primary results, LaViLa++, pretrained on the state-of-the-art model LaViLa shows a great generalization on other public benchmarks in Figure 1. For instance, it is essential to note that our method competes with state-of-the-art models like HelpingHands [1] and HENASY [2] in Table 1 on EK-100-MIR, which use extensive grounding data and more parameters for training. Our LaViLa++ also improves the action recognition of 3805 classes than LaViLa on Epic-Kitchen-100 by +4.53%. Besides, LaViLa++ also achieves a human level of temporal understanding on ActionBench. This underscores the significant impact of EgoNCE++.
> - **Overfitting Concerns**: Overfitting appears unlikely, as we observe minimal descreases across most out-of-domain benchmarks, such as  Epic-Kitchens, CharadesEgo, and EGTEA. Also, the diversity of video content in Ego4D and the variability introduced by LLM-based text generation contribute to preventing overfitting. We include vocabulary frequency statistics to support the vocabulary diversity provided by LLM in table below:
> | Training Data Vocabulary | Verb  | noun |
> |----------------|-------|------|
> | positives       | 7679  | 2171 |
> | positives+negatives  | 15643 | 6654 |
> | Δ (by LLM) | 7964  | 4483 |

---

> ### Author Response · Authors · 2024-11-22
> **Official Comment to Reviewer 3po1‘s Questions [2/2]**
>
> **Q1: Comparison of the Proposed Asymmetric Contrastive Objective with Other Contrastive Techniques in Video-Language Modeling**
>
> Table 5 records existing hard negative mining approaches in egocentric contexts, but these EgoVLMs fail to address the specific challenges posed by EgoHOIBench. Our method, incorporating asymmetric V2T and T2V losses, effectively addresses these gaps and leads to improved performance on challenging downstream tasks, as shown in Figure 1.
>
> **Q2: Can We Elaborate on the Robustness in Generated Negative Captions and How the Bias in LLM Affects the Model's Performance?**
>
> We appreciate the suggestion to evaluate the robustness of the generated captions and the potential bias in LLMs.
> - **Elimination of False Negatives**: During pretraining, we remove the false negatives by using spaCy for word extraction and the Ego4D dictionary for validation. In EgoHOIBench, we also rely on the semantic understanding of LLMs to ensure the generated negatives differ significantly in meaning. We explicitly prompt LLMs to "make the action in the text significantly different while keeping the other words unchanged."
> - **Bias in LLM When Constructing EgoHOIBench by other LLMs**: To further evaluate potential biases in LLMs, we generated EgoHOIBench using alternative LLMs such as DeepSeek-200B and GPT-4o. Results in table below show that LaViLa++ and EgoVLP++, pretrained using EgoNCE++ and LLaMA generated negatives, consistently improves HOI-verb and HOI-noun recognition. The results are weaker than those on LLaMA-EgoHOIBench due to the vocabulary shift. The numbers in the table denotes verb/noun/action accuracy respectively:
>
> | Method | LLaMA-EgoHOIBench| GPT-4o-EgoHOIBench| DeepSeek-EgoHOIBench |
> |---|---|---|---|
> | LaViLa| 46.61 / 74.33 / 36.85 | 45.18 / 72.82 / 38.20 | 44.16 / 62.28 / 34.09|
> | LaViLa++ | 80.63 / 75.30 / 63.17 | 53.95 / 73.65 / 44.63 | 52.73 / 63.48 / 39.15|
> | EgoVLP  | 40.27 / 68.60 / 30.16 | 41.44 / 66.55 / 33.96 | 40.75 / 56.42 / 30.56 |
> | EgoVLP++| 56.11 / 69.05 / 41.63 | 50.89 / 70.19 / 41.78 | 50.59 / 60.84 / 38.04|
>
>  The GPT-4o and DeepSeek tend to generate different words compared with LLaMA, whose differences are shown in the table below. In the table below, we calculate vocabulary statistics, where the value in position (i, j) represents the proportion of the vocabulary from the i-th set that overlaps with the j-th set:
>
> |           | LLaMA-EgoHOIBench | GPT-4O-EgoHOIBench | DeepSeek-EgoHOIBench |
> |---------------------------|-------------------|--------------------|---------------------------|
> | **LLaMA-EgoHOIBench**         | 100%              |   37.87%           | 41.46%                    |
> | **GPT-4O-EgoHOIBench**        | 37.87%            | 100%               | 58.04%                    |
> |**DeepSeek-EgoHOIBench** | 41.46%            | 58.04%             | 100%                      |
>
>
> - **Vocab-based Negative Mining is Competitive with LLM-based Method**: We mitigat potential biases from LLMs by using a vocab-based approach for caption augmentation. This strategy reduces over-reliance on pre-trained LLMs and maintains diversity of negative samples, which helps avoid bias while still generating effective negatives.
>
> **Q3: How Does EgoHOIBench Compare to Existing Datasets, and the Limitations When Applied to Real-World Scenarios?**
>
> - EgoHOIBench is designed to capture a wide range of human activities from the diverse Ego4D dataset, making it a valuable complement to existing benchmarks that focus on narrower scenarios (e.g., Epic-Kitchens, which is limited to kitchen scenarios). Our results on EgoHOIBench demonstrate its potential to improve general egocentric hand-object interaction recognition. Additionally, diverse datasets like Epic-Kitchens and EGTEA demonstrate the generalization ability of our pretrained models to domain-specific scenarios, such as kitchen environments.
>
> We hope that our responses have adequately addressed your concerns.  Should you have any remaining points or further steps we can take to strengthen our work, we would greatly appreciate your continued feedback. Thank you again for your valuable feedback.
>
> References:
>
> [1] Helping hands: An object-aware ego-centric video recognition model, ICCV2023.
>
> [2] Henasy: Learning to assemble scene-entities for interpretable egocentric video-language mode, NeurIPS2024
>
> [3] Improving CLIP Training with Language Rewrites, NeurIPS2023

---

> ### Author Response · Authors · 2024-11-25
> **Kindly reminder: The rebuttal deadline approches (November 26th AoE)**
>
> Dear reviewer 3po1,
>
> We again thank you for your valuable feedback, which has greatly helped improve the quality of our paper. As the rebuttal deadline approaches (November 26th AoE), we kindly ask if you could review our rebuttal to see whether it addresses your concerns.
>
> We would greatly appreciate any additional feedback or raise further questions you may have during this discussion period.
>
> Best regards,
>
> Authors

---

> > ### Comment · Reviewer_3po1 · 2024-11-26
> >
> > 1. The comparison between the proposed method and the existing technique was presented in the supplement. The author should point to the supplement since it is one of the main contributions.
> > Also, the difference in computation resources should be stated in the revision.
> >
> > 2. In demonstrating the model generalizability, the author showed performance gain on other datasets in recognition tasks. However, it is unclear why such a method doesn't contribute to performance gain in benchmarks such as Epic-Kitchens, CharadesEgo, and EGTEA.
> >
> > 3. Regarding the bias, the author showed results across the use of various LLMs and showed constant improvement.
> >
> > Given the explanation, I would raise my score accordingly.

---

> > > ### Author Response · Authors · 2024-11-26
> > > **Official Comment to Reviewer 3po1‘s Comments**
> > >
> > > Thank you for your constructive suggestions on our paper and for improving the score to acknowledge our efforts. We address your additional comments as follows:
> > >
> > > **C1: Suggestion on the manuscript**
> > >
> > > Thank you for your valuable feedback. We will revise our paper accordingly before the revision deadline.
> > >
> > > **C2: Why our method doesn't contribute to performance gain in benchmarks such as Epic-Kitchens, CharadesEgo, and EGTEA?**
> > >
> > > We believe there may be a misunderstanding caused by our inadequate explanation. Regarding the generalization on benchmarks such as Epic-Kitchens, EGTEA and CharadesEgo, our method achieves:
> > > - **Notable Improvements under the Zero-Shot Setting Comparing LaViLa++ and LaViLa (Figure 5 and Table 1)**: As shown in Figure 5, the LaViLa++, pretrained by EgoNCE++, demonstrates consistent performance improvements across most benchmarks, including Epic-Kitchens (on both the multi-instance retrieval task and action recognition task, referred to as EK-100-MIR and EK-100-CLS, respectively), CharadesEgo, and EGTEA. For instance, the zero-shot performance gains of LaViLa++ compared to LaViLa on these benchmarks are detailed below:
> > >
> > > |    Zero-Shot Evaluation| Epic-Kitchens-100-MIR | Epic-Kitchens-100-MIR | Epic-Kitchens-100-CLS |   EGTEA  |   EGTEA  | CharadesEgo |
> > > |:--------:|:---------------------:|:---------------------:|:---------------------:|:--------:|:--------:|:-----------:|
> > > |     metric     |          mAP          |          nDCG         |        top1-acc       | mean-acc | top1-acc |     mAP     |
> > > |  LaViLa  |         30.8         |         32.0|         10.14         |   30.9  |   35.1   |     20.6    |
> > > | LaViLa++ |         32.0|          33.1|         14.67         |   34.0  |   35.4   |     20.9    |
> > >
> > > - **Consistent Improvements Across EgoVLMs on Various Benchmarks (Table 7 in Supplementary)**: A detailed comparison of EgoVLMs pretrained using EgoNCE++ is provided in Table 7 of the supplementary material. This comparison highlights consistent improvements across multiple models, including EgoVLP, EgoVLPv2, and LaViLa, on various benchmarks like Epic-Kitchens.
> > >
> > >
> > > We hope that our responses have adequately addressed your concerns. We would greatly welcome any additional feedback or raise further questions you may have during this discussion period.

---

### Official Review · Reviewer_7uyU · 2024-11-01

**Soundness:** 2
**Presentation:** 2
**Contribution:** 2
**Rating:** 8
**Confidence:** 5

**Summary:**

In this paper, the authors propose a new dataset, named EgoHOIBench, which consists of multiple choice questions for video clips in which there are two settings for verb and noun understanding. There are 10 choices for each case in which the verb or the noun respectively have been changed to create a hard negative. The paper finds that current Egocentric VLMs are unable to handle this change and so a new loss, named EgoNCE++ in which an asymmetric objective forces the model to understand minor differences in text yet for video groups the representations via their noun representations. The results show that for the EgoHOIBench, training current Egocentric VLMs with the new objective leads to an improvement in performance as well as for other downstream tasks.

**Strengths:**

* The results of the different Egocentric VLMs on downstream tasks after being trained with the new EgoNCE++ objective are good with nice increases in performance.
* Creating and proposing an asymmetric loss with EgoNCE++ is interesting and makes a lot of sense in how these two settings need to be treated differently whereas in the past this has not necessarily been true.
* There are a lot of results (and qualitative figures within the appendix) to showcase the method and struggles of the current methods without the EgoNCE++ objective.

**Weaknesses:**

# Weaknesses
* One potential reason for the new loss doing so well on the constructed HOI-Bench is as the benchmark has been designed in the same way as the loss function. A model that has been trained using a loss with negatives that represent the same style of negatives that are in the ground truth answers is certainly going to do better.
* Models seem to already do well on noun focused tasks, so it isn't clear to me why there is a large focus on still clustering videos based on similar noun representations only.
* Currently, the method section (Section 3) is lacking some important information regarding how the negatives are generated beyond an LLM/using the vocabulary of the dataset. Also, if there is any checking that is done to reduce/remove false negatives.
* The details of how HoI-bench is collected within the main paper is very scarce. The appendix includes some more information, but is still quite lacking. Details of why the number of videos were chosen, number of captions, any choice of category within Ego4D etc. is missing
* Instead of defining a new metric to look at the positive/negative similarities within equation 6/figure 7. A histogram could have been used instead which might have given a clearer picture (again using the max negative). It's less intuitively clear from the figure what the numbers represent, especially as these have been multiplied by 100, emphasising the small differences even further.

# Additional Comments
Table 3 is inconsistent with how it uses lower case and upper case compared to the rest of the paper. Additionally, 'ours' is used here instead of EgoNCE++.

**Questions:**

1. Has an investigation/analysis been carried out regarding the fact that the loss is designed in the same way as the answers within the dataset? An outcome of this can be seen within Table 4 perhaps, where the choice of generator leads to a large increase in results for HOIBench, but is marginal for the EK-100-MIR task.
2. What does it mean by the sentence not made excessively difficult? Is this because false negatives could be introduced via semantic matching? If so, what is used to prevent this?
3. Why cluster videos based on similar noun representations only, instead of a mix of verb and noun representations?
4. How are negatives introduced into the video-to-text loss to ensure that false negatives are not included? It is mentioned that either vocabulary from the dataset or an LLM is used to generate the negatives, but there is no information on how this is done. Is this the same as the collection information for HOI-Bench
5. Were all videos chosen for HOI-Bench from the validation set in a similar fashion to EgoMCQ? Or were some videos excluded? How was 10 settled on as the number of captions?
6. When constructing the dataset, has any care been taken to ensure that the LLM used to generate the questions isn't hallucinating? Were any measures put in place to remove noise and ensure a cleaner outcome for the dataset? Also, has any human checking/evaluation been carried out to get a sense of how clean the data is?
7. Has an ablation been carried out in which EgoNCE++ (ours in Table 3) is used for the loss and InfoNCE is used for the V2T loss?
8. For figure 6, has this been evaluated with how the number of negatives scales for the base models without the EgoNCE++ loss? It would be interesting to see how this compares for both models.
9. Has a histogram of positive/negative similarities been created instead of using the new PND metric in Figure 7?
10. It would be interesting to see Figure 3 after the EgoNCE++ objective has been applied if this has been created to see how this differs.

---

> ### Author Response · Authors · 2024-11-22
> **Official Comment to Reviewer 7uyU's Weaknesses and Comments [1/2]**
>
> We greatly appreciate the time and effort you invested in providing these detailed observations, questions, and comments. We have carefully considered your comments and outlined our responses and proposed revisions below. We hope these clarifications and revisions address your concerns and further enhance the manuscript. Note that we refer to the table/figure numbers in our revised paper.
>
> **W1: Similar Design of Loss and EgoHOIBench**
>
> We acknowledge that the similarity between the benchmark design and the loss function may contribute to large improvement on EgoHOIBench. However, our designed loss also performs well on other benchmarks. For example, as shown in Table 1, our method competes with SoTA models like HelpingHands [1] and HENASY [2] which use extensive grounding data and more parameters for training. These results highlight the meaningful impact of EgoNCE++ even without external datasets.
>
>
> **W2: Why There Is a Large Focus on Still Clustering Videos Based on Similar Noun Representations Only?**
>
> Since our V2T negative mining on HOI-verbs might damage the strong recognition towards noun, we aim to maintain the noun clustering nature by T2V positive sampling on nouns. Relavent ablation shows its necessity in Table 12 in supplementary material, clustering video representations based solely on nouns yields the largest gain.
>
>
> **W3: Details on Negative Generation (Section 3) and Reduction of False Negatives**
>
> - **Details on Negative Generation**: Thank you for your suggestion of clarifying how negatives are generated in Section 3. In the revised version of our manuscript, we provide more details about how training negatives are constructed using LLMs or dataset vocabulary: (1) LLM-based. Following a process similar to EgoHOIBench, we prompt LLM with in-context learning examples to generate Json-formatted sentences that are semantically distinct from the original text. (2) Vocab-based. Using spaCy, we extract all verbs and nouns from the pretraining datase and replace HOI-nouns or HOI-verbs with randomly selected extracted words.
> - **Reduction of False Negative**: During the negative generation process for both methods, we leverage the established synonym list from Ego4D to ensure minimal overlap and eliminate false negatives, as stated in Line882-885. The LLM-based method would be more capable of identifying synonyms due to the powerful semantic understanding capability, while the vocab-based method is only aware of the synonyms in Ego4D vocabulary.
>
> **W4: Details on HOI-Bench Data Collection**
>
> We acknowledge the need for clearer information regarding the construction of EgoHOIBench. The main paper will be updated to explain the rationale behind the number of videos, captions, and category choices from Ego4D. Specifically:
> - **Number of Negatives - 10**: Setting the number of negatives to 10 (i.e. 10 noun negatives, 10 verb negatives) will form 100 HOI negatives, which aligns with the typical action recognition setting. Additionally, we aim to examine the HOI recognition capabilities of EgoVLMs without requiring a large number of classes. You can download our data from our anonymous code repository.
> - **The Chosen Number of Videos - 29K**: To filter the videos from EgoMCQ that feature hand-object interactions,  we applied two filtters: (1) Captions starting with #C (indicating the wearer, thus excluding data involving others, which starts with #O), and (2) Captions that contain both verbs and nouns, excluding non-interactive daily activities (e.g., '#C walks away'). As a result, the video captions primarily follow the format '#C C #HOI-verb #HOI-noun ...', where '...' represents other phrases or articles, as our statement in Line862-864. Such filtering process resulted in 29K videos.
> - **Choice of Category**: We did not perform special selection of categories during the video filtering process. The vocabulary statistics of EgoHOIBench is shown in Figure 8 in supplementary.
>
> **W5: Metric Clarification (Original Equation 6, Now Removed)**
>
> Thank you for suggesting drawing a histogram instead. We follow this suggestion in the revised version and present a histogram of positive and negative similarities in Figure 6. This visual representation offers a more intuitive understanding of distribution shift, where our EgoNCE++ effectively makes video-positives better matched than video-negatives.
>
>
> **Comment1: Inconsistent Lower and Upper Cases in Table 3**
>
> Thanks for pointing it out, we have revised it in the updated version.

---

> ### Author Response · Authors · 2024-11-22
> **Official Comment to Reviewer 7uyU's Questions [2/2]**
>
> **Q1: Analysis on Loss and Dataset Design**
>
> We haven't analyze much about the relation between our loss and EgoHOIBench. Instead, to evaluate potential biases in LLMs, we generated EgoHOIBench using alternative LLMs such as DeepSeek-200B and GPT-4o. Results in table below show that LaViLa++, pretrained using EgoNCE++ and LLaMA generated negatives, consistently improves HOI-verb and HOI-noun recognition. The results are weaker than those on LLaMA-EgoHOIBench due to the vocabulary shift. The numbers in the table denotes verb/noun/action accuracy respectively:
>
> | Method | LLaMA-EgoHOIBench| GPT-4o-EgoHOIBench| DeepSeek-EgoHOIBench |
> |---|---|---|---|
> |LaViLa|46.61 / 74.33 / 36.85|45.18 / 72.82 / 38.20|44.16 / 62.28 / 34.09|
> |LaViLa++| 80.63 / 75.30 / 63.17|53.95 / 73.65 / 44.63|52.73 / 63.48 / 39.15|
> |EgoVLP| 40.27 / 68.60 / 30.16|41.44 / 66.55 / 33.96|40.75 / 56.42 / 30.56|
> |EgoVLP++|56.11 / 69.05 / 41.63 |50.89 / 70.19 / 41.78 |50.59 / 60.84 / 38.04|
>
> Besides, the LaViLa++ that pretrained on EgoNCE++ not only performs well on EgoHOIBench, but also generalizes to other benchmarks. For example, as shown in Table 1, our method competes with SoTA models like HelpingHands [1] and HENASY [2]. Considering external datasets and parameters used by [1][2], the improvement is significantly more than marginal. As for the results in Table 4, to ensure fair comparison for all ablations, we employ a fixed number of 10 negatives on EgoVLP. However, our primary results explore more on LaViLa, as shown in Figure 1 and Figure 7, resulting in larger gains than those on EgoVLP.
>
> **Q2: Clarification on “Not Made Excessively Difficult” Sentence**
>
> The term "difficult" refers to the fine-grained nature of negative examples, which differ by only one word from the positive captions. Our aim is to examine the HOI understanding capabilities of EgoVLMs without requiring  a very large number of classes like those in Epic-Kitchens (3805). To avoid false negatives, we prompt LLM to generate negatives with only one word difference, but with drastically different semantics (not made excessively difficult). The LLM is instructed to "make the action in the text significantly different while keeping other words unchanged." Detailed prompt examples can be found in Figure 13 in the supplementary material.
>
> **Q3: Noun-Based Clustering Only**
>
> Please refer to our response in W2.
>
> **Q4: Prevention of False Negatives in Video-to-Text Loss**
>
> Please refer to our response in W3.
>
> **Q5: Data Selection for HOI-Bench**
>
> The video-text data in EgoHOIBench are chosen from EgoMCQ. Please refer to our detailed response in W4.
>
> **Q6: Mitigation of Hallucination During Dataset Construction**
>
> To minimize hallucinations and ensure a consistent output format, we employ in-context learning and restrict the model's responses to Json format. We randomly sampled 200 data samples for manual evaluation and found no hallucinations.
>
> **Q7: Ablation with EgoNCE++ for Loss and InfoNCE for V2T**
>
> We conducted ablation studies using EgoNCE++ for T2V loss and InfoNCE for V2T loss. The results in Table 3 show that the model continues to benefit from our T2V positive sampling loss:
>
> | V2T| T2V|verb| noun | action | avg.mAP  | avg.nDCG|
> |---|---|---|---|---|---|---|
> | InfoNCE| InfoNCE| 40.70 | 68.86 | 30.51  | 22.1| 26.5|
> | InfoNCE|EgoNCE++| 40.60 | 69.15 | 30.62  | 22.3 | 26.7 |
> | EgoNCE++|InfoNCE | 54.56 | 68.96 | 40.31  | 22.4  | 26.7 |
> | EgoNCE++ | EgoNCE++ | 56.11 | 69.05 | 41.63  | 22.7  | 27.1|
>
> **Q8: Impact of Number of Negatives on Base Models**
>
> We have tested the effect of applying the same number of 10 negatives without EgoNCE++ (rule-based method, defined in line 461-463) and found limitations of rule-based method in generalization, as shown in Table 4. The limited generalization of rule-based methods may result from having seen the sentences during pretraining, while the EgoNCE++ generates new fine-grained negative options that are never seen during previous training processes.
>
> **Q9: Histogram for Positive/Negative Similarities**
>
> Please refer to our response in W5.
>
> **Q10: Feature Space Comparison After EgoNCE++**
>
> Thank you for this insightful suggestion. We have visualized feature spaces after applying EgoNCE++ and included these in the supplementary materials. Combining the visualization of feature spaces and the video-text matching similarity scores, the results show that while the overall clustering remains object-centric, video-text matching improves, confirming the effectiveness of our method.
>
> We hope our clarifications address your questions. Should you have any remaining points or further steps we can take to strengthen our work, we would greatly appreciate your continued feedback. Thank you again for your constructive and valuable suggestions.
>
>
> References:
>
> [1] Helping hands: An object-aware ego-centric video recognition model, ICCV2023.
>
> [2] Henasy: Learning to assemble scene-entities for interpretable egocentric video-language mode, NeurIPS2024

---

> > ### Comment · Reviewer_7uyU · 2024-11-25
> >
> > Thank you for providing responses and answering my questions from my initial review. Because of this detailed response and updates to the paper, I will increase my score to accept.

---

> ### Author Response · Authors · 2024-11-25
> **Grateful to Reviewer 7uyU for the Constructive Feedback and Appreciation**
>
> Dear reviewer 7uyU,
>
> We sincerely appreciate your acknowledgment of our efforts in addressing your concerns. Your constructive and insightful suggestions have been critical in enhancing the the quality of our work. We are also grateful for the revised score—thank you for recognizing our improvements.
>
> We welcome any further discussion and are glad to provide additional clarification if needed.
>
> Best regards,
>
> Authors

---

### Official Review · Reviewer_DGPT · 2024-11-07

**Soundness:** 4
**Presentation:** 4
**Contribution:** 3
**Rating:** 8
**Confidence:** 3

**Summary:**

This paper introduces a novel way to identify HOIs in EgoVLMs, which addresses the limitation of current egocentric models regarding verb recognition. They propose EgoHOIBench, a new benchmark designed to evaluate understanding in EgoVLMs, that can evaluate EgoVLM’s capabilities in understanding variations of HOI combinations. In addition, their experiment demonstrates a stronger robustness towards recognizing nouns through their analysis of EgoHOIBench performance on HOI-verbs and HOI-nouns. Furthermore, they propose EgoNCE++, an asymmetric contrastive learning objective to address these limitations by enhancing model robustness in handling fine-grained verb and noun variations within HOIs for egocentric video-language pretraining, which successfully fulfills their aim to preserve the object-centric nature of the feature space without additional visual data usage or architectural changes.

**Strengths:**

•	The paper pinpoints the current weakness that existing benchmarks in egocentric vision with EgoHOI are limited. While some of them are only emphasized in kitchen scenarios, the others fail to provide effective supervision for understanding the nuances of HOI combinations that make our current model “a lack of fine-grained negative supervision during pretrained process”.

•	The author discovers this critical gap and gives out their own solution in making EgoHOIBench, a benchmark specifically designed to test HOI comprehension in egocentric contexts, which is designed to more effectively evaluate the ability of EgoVLMs to select the correct sentence from multiple HOI-related options using video-text matching.

•	Compared with InfoNCE and EgoNCE, which often sample easy negative pairs without employing hard negative mining for text, and distinguish EgoHOIs based on verb-noun variation, EgoNCE++ incorporates asymmetric video-to-text and text-to-video losses, which enables the model to better understand HOI combinations by generating negatives through HOI-related word changes and preserves object-centric feature properties by clustering video representations based on similar nouns.

•	The paper is well-written and also the experiments are well-structured by covering a large range of benchmarks in Egocentric Vision tasks across commonly used datasets: Open-vocabulary recognition, multi-instance retrieval, and action recognition.

**Weaknesses:**

•	The paper introduces EgoNCE++ as an asymmetric contrastive learning objective. Figure 3 shows the visualization of LaViLa’s feature space indicating both video and text feature space exhibit the object-centric property and suggests that video-noun matching is easier than video-verb matching since noun-anchored embeddings form tighter clusters, while verb-anchored embeddings are more dispersed. Can we make a feature space comparison after applying EgoNCE++ to see if the verb-anchored changed?

•	The paper mentions that it utilizes the LLaMA-3-8B model to generate HOI candidates through in-context learning. It would be better if we could also conduct additional studies on the performance gains versus computational expense(FLOPs) in order to better know the trade-off for the approach.

•	EgoNCE++ works significantly well on the author’s self-designed benchmarks EgoHOIBench, but it seems that for other datasets the improvement is marginal: EK100-MIR, CharadesEgo shown in Figure.5; EK-100-MIR and EGTEA in Table 9 for zero-shot setting; LaViLa and LaViLa++ in Table 10 and Table 7. Will there be a possible overfitting or lack of variety since the primary performance measure is based on EgoHOIBench, which specifically focuses on nuanced HOI comprehension?

**Questions:**

Questions are written with weakness.

---

> ### Author Response · Authors · 2024-11-22
> **Official Comment to Reviewer DGPT**
>
> Thank you for appreciating our work and for providing thoughtful feedback that helps us further improve our work. Note that we refer to the table/figure numbers in our revised paper. Below, we address your concerns in detail:
>
> **W1: Can We Make a Feature Space Comparison after Applying EgoNCE++ to See If the Verb-Anchored Changed?**
>
> We appreciate the insightful suggestion to include feature space comparisons after applying EgoNCE++. To address this, we have provided additional visualizations in Figure 13 of the Supplementary Material. These visualizations reveal that the feature space preserves its object-centric nature. Notably, the inherent structure of the video or text representation space does not necessarily impact the video-text matching results. Our findings, illustrated in Figure 5 (b) and Figure 6, demonstrate that our approach enhances video-text matching by making the correct video-text pair more distinguishable from negative video-text ones.
>
> **W2: Can We Conduct Additional Studies on the Performance Gains versus Computational Expense (FLOPs) in Order to Better Know the Trade-Off for the Approach?**
>
> Thank you for highlighting the computational trade-offs of our approach. Our current approach, which uses LLaMA-3-8B to generate HOI candidates, operates offline and requires a higher computational cost. Specifically, generating training data for 2.5M samples required approximately 192 GPU hours on A6000 hardware (equivalent to 276.48 ms per 10 negatives). In comparison, a vocab-based negative sampling method operates far more efficiently, requiring only 0.039 hours on 16 CPU threads (equivalent to 0.05616 ms per 10 negatives) and enabling online updates. A detailed comparison of the trade-offs between the LLM- and Vocab-based methods on EgoVLP are presented in the table here:
>
> | Method | Speed    | EgoHOIBench | EK100-MIR |
> |---------------|---------------------|-------------|-----------|
> | LLM-based  | 276.48 ms/10 neg  | 41.63 | 22.7      |
> | Vocab-based| 0.05616 ms/10 neg | 40.07| 22.5      |
>
> **W3:   It Seems That for Other Datasets the Improvement of EgoNCE++ Is Marginal. Will There Be a Possible Overfitting or Lack ff Variety Since the Primary Performance Measure Is Based on EgoHOIBench?**
>
> We recognize that while EgoNCE++ achieves significant gains on EgoHOIBench, its improvements appear relatively modest on datasets such as CharadesEgo. Below, we present a detailed analysis of the factors contributing to these less significant gains:
>
> - **Notable Improvement under the Same Setting Comparing LaViLa++ and LaViLa (Figure 5 and Table 1)**: Notably, our method EgoNCE++ brings large improvement under the same experimental setup. For example, LaViLa++ improves action recognition across 3805 classes on Epic-Kitchen-100 by +4.53%, and achieves a human level of temporal understanding on ActionBench. Our improvements on EK-100-MIR are comparable to HelpingHands [1] and HENASY [2]  (see Table 1 on EK-100-MIR), which benefits from extensive grounding data and larger parameter sizes used by our competitors.
>
>  - **Overfitting Concerns**: We find no significant decreases in benchmark metrics, suggesting overfitting is unlikely. The diversity of video content in Ego4D and the variability introduced by LLM-based text generation contribute to preventing overfitting. To illustrate this, we provide vocabulary frequency statistics in table here:
>
> | Training Data Vocabulary | Verb | noun |
> |---|---|---|
> | positives   | 7679  | 2171 |
> | positives+negatives  | 15643 | 6654 |
> | Δ (by LLM) | 7964|4483 |
>
> - **Large Domain Gaps in Zero-Shot Setting on Datasets like CharadesEgo (Table 7)**: The marginal generalization improvement on datasets like CharadesEgo can be attributed to the out-of-distribution (OOD) gap between Ego4D and CharadesEgo, which is a common sense in egocentric domain as posed by EgoVLP[3].
>
> - **Impact of LoRA Fine-tuning in Fine-tuning Settings (Table 10)**: The performance on EK100MIR and EGTEA in fine-tuning settings is influenced heavily by the base model's capabilities. Our LoRA fine-tuning approach limits significant changes in feature space distribution, therefore resulting in limited improvements as depicted in Table 10.
>
> - **Stability When Pretraining on Larger Model (Table 9)**: While EgoNCE++ consistently boosts performance, larger models like LaViLa-Large already possess a strong baseline for comprehension. Consequently, the relative gains from negative text supervision alone may become relatively smaller when pretrained on larger models.
>
> We hope these clarifications address your questions. Please let us know if there are additional concerns or suggestions. We greatly appreciate your constructive feedback and support.
>
> References:
>
> [1] Helping hands: An object-aware ego-centric video recognition model, ICCV2023.
>
> [2] Henasy: Learning to assemble scene-entities for interpretable egocentric video-language mode, NeurIPS2024.
>
> [3] Egocentric Video-Language Pretraining, NeurIPS2022.

---

> ### Author Response · Authors · 2024-11-25
> **Kindly reminder: The rebuttal deadline approches (November 26th AoE)**
>
> Dear reviewer DGPT,
>
> We again thank you for your valuable feedback, which has greatly helped improve the quality of our paper. As the rebuttal deadline approaches (November 26th AoE), we kindly ask if you could review our rebuttal to see whether it addresses your concerns.
>
> We would greatly appreciate any additional feedback or raise further questions you may have during this discussion period.
>
> Best regards,
>
> Authors

---

> ### Comment · Reviewer_DGPT · 2024-11-26
>
> I think the author properly answered my question during the rebuttal period.
> • These visualizations in Fig.12 did reveal that the feature space preserves its object-centric nature.
> • The additional FLOPs experiments make sense and do not significantly increase the computational resource based on LLMs usage.
> • I agree with authors that after the comparison, the model did helps to other benchmark not including their own EgoHOIBench.
>
> Given the information above, I will raise my point and thanks for the rebuttal and comments made by authors.

---

> > ### Author Response · Authors · 2024-11-26
> > **Appreciation to Reviewer DGPT for Valuable Feedback and Recognition**
> >
> > Dear reviewer DGPT,
> >
> > We sincerely appreciate your recognition of our efforts to address your concerns. Your constructive and insightful suggestions have been instrumental in improving the quality of our work. We are also grateful for the revised score and thank you for acknowledging our progress.
> >
> > We welcome any further discussion and are glad to provide additional clarification if needed.
> >
> > Best regards,
> >
> > Authors

---

### Official Review · Reviewer_Ns3f · 2024-11-09

**Soundness:** 3
**Presentation:** 3
**Contribution:** 3
**Rating:** 6
**Confidence:** 5

**Summary:**

This paper focuses on understanding hand-object interactions using egocentric video-language models (EgoVLMs). The authors introduce a new benchmark called EgoHOIBench to evaluate EgoVLMs' ability to distinguish between similar HOI descriptions by changing verbs or nouns. They then propose EgoNCE++, an asymmetric contrastive learning objective to improve the models' sensitivity to nuanced changes in HOI-related language by using LLM-generated negative samples to enhance text supervision and preserve object-centric video representations. The paper shows that EgoVLMs generally behave better at object recognition while struggling with action. The experiments show that EgoNCE++ enhances performance across three EgoVLMs and improves generalization on seven downstream EgoHOI tasks.

**Strengths:**

1. The paper is clear and well-motivated. The authors first diagnose current EgoVLMs' capability on the proposed EgoHOIBench benchmark and then propose a method to address the problem. This makes the paper easy to follow and provides insights to readers.
2. The proposed EgoHOIBench provides a targeted evaluation for current egocentric video-language models regarding the capability of understanding hand-object interactions with variations in verbs and nouns. The authors also provide an inspiring analysis of the current EgoVLM and find their common failure to understand actions.
3.  The paper conducts extensive experiments and shows the proposed EgoNCE++ consistently improves model performance across various state-of-the-art EgoVLMs, highlighting its versatility and effectiveness.
4. The authors release the codes that promote reproducibility.

**Weaknesses:**

The major concern about this paper is its novelty. The idea of strengthening fine-grained compositional understanding by constructing hard-negative examples is not novel e.g. (Yuksekgonul et al., 2023). What are the major differences between the proposed method and previous works?

**Questions:**

Please see weakness section for the questions.

---

> ### Author Response · Authors · 2024-11-22
> **Official Comment to Reviewer Ns3f**
>
> We appreciate the reviewer’s thoughtful assessment and acknowledgment of our paper's clarity, motivation, and contributions. Below, we address the concerns in detail:
>
> **W1: Response to Concerns about Novelty**
>
> We understand the reviewer's interest in better distinguishing the novelty of our approach, particularly in relation to existing works such as negCLIP [1]. Below, we provide a detailed response to emphasize the unique contributions of our work:
>   - **Distinct Problems Addressed**: While both negCLIP and our work focus on compositional understanding, the core problems each approach tackles are different. Specifically, negCLIP primarily addresses challenges related to positional and word order sensitivity, such as errors exemplified by statements like "the grass is eating the horse". In contrast, our work focuses on HOI understanding with a particular emphasis on distinguishing precise verbs or nouns in free-form language queries, such as differentiating "C drops the paper" from other similar actions like "C {other verbs} the paper".
>   - **Unique Negative Construction Process**: Unlike negCLIP, which relies on hand-crafted text shuffling, our approach leverages advanced LLMs or a more flexible and efficient vocab-based method (as detailed in Section 3) to generate negatives.
>   - **Novel Asymmetric Loss Design for EgoHOI**: Our EgoNCE++ introduces an innovative asymmetric loss structure specifically designed for egocentric data， which unifies vision-to-text (V2T) negative mining and text-to-vision (T2V) positive sampling within a contrastive loss framework. This design effectively addresses the unique challenges posed by egocentric video data.
>   - **Improved Generalization**: Unlike previous works such as negCLIP, which sometimes exhibits reduced generalization compared to CLIP, our approach maintains robust performance for EgoVLMs without such degradation. Such generalization is likely due to the intrinsic similarity of egocentric videos (featuring hand-object interactions) compared to the broader variance observed in third-person datasets.
>
> We hope these clarifications sufficiently address the novelty and unique contributions of our work. Should you have additional questions or concerns, we would be happy to provide further explanatioins. Thank you again for your thorough review and constructive feedback.
>
> References:
>
> [1] When and why Vision-Language Models behave like  Bags-of-Words, and what to do about it? ICLR2023

---

> > ### Comment · Reviewer_Ns3f · 2024-12-02
> >
> > Thanks to the authors for the responses about the novelty. I think the comparison with negCLIP shows the contribution of this paper and I'd like to keep my rating as boderline accept.

---

> ### Author Response · Authors · 2024-11-25
> **Kindly reminder: The rebuttal deadline approches (November 26th AoE)**
>
> Dear reviewer Ns3f,
>
> We again thank you for your valuable feedback, which has greatly helped improve the quality of our paper. As the rebuttal deadline approaches (November 26th AoE), we kindly ask if you could review our rebuttal to see whether it addresses your concerns.
>
> We would greatly appreciate any additional feedback or raise further questions you may have during this discussion period.
>
> Best regards,
>
> Authors

---

### Meta-Review · Area_Chair_4Q8r · 2024-12-25

**Metareview:**

The submission addresses the problem of understanding hand-object interactions in egocentric videos. It introduces an asymmetric contrastive learning objective whose goal is to handle fine-grained verb and noun variations when describing hand-object interactions in natural language. It also introduces a new benchmark "EgoHOIBench" to evaluate video-language models' understanding of fine-grained human-object interactions. The proposed approach achieves competitive performance on multiple benchmarks. After rebuttal, the submission received two accepts (8) and two borderline accepts (6). Most of the questions were addressed by the rebuttal. The AC finds no basis to overturn the consensus reached by all reviewers, and thus recommends acceptance.

**Additional Comments On Reviewer Discussion:**

There were novelty concerns raised by reviewers Ns3f and 3po1, which were adequately addressed by the authors. The authors should incorporate the reviewers' suggestions in the final version of their submission.

---

### Decision · Program_Chairs · 2025-01-22

Accept (Poster)